# MoLE-GNN: Parameter-Efficient Fine-Tuning of Graph Neural Networks with Mixture-of-Experts

## Abstract

Graph Neural Networks (GNNs) are gaining popularity for modeling non-Euclidean data due to their ability to capture local and global structure using message-passing techniques. In real-world scenarios, such as graph classification task, the size of graphs within the same dataset can vary significantly. This warrants an investigation into *depth-sensitivity* of graphs, leading to selection of optimal number of GNN layers according to the size of the graph. Traditional GNNs suffer from a static choice of number of layers for the graphs as it leads to underfitting in the large graphs and overfitting in the smaller ones. Although recent Mixture-of-Experts (MoE) GNN models solve this problem by adaptively selecting depth-sensitive expert networks, they have high computational and memory overhead. To overcome these challenges, we introduce a new hybrid model named MoLE-GNN that combines parameter-efficient adapter modules with GNN experts, supporting dynamic expert assignment with minimal fine-tuning. It drastically minimizes trainable parameters (tunes only 5.1% of the total parameters) and improves generalization, particularly in low-resource environments. Our extensive experiments across inductive, transductive, and link prediction tasks demonstrate that MoLE-GNN consistently outperforms both full fine-tuning and state-of-the-art PEFT baselines, offering a scalable and effective approach for fine-tuning GNNs on diverse graph topologies. Moreover, MoLE-GNN surpasses existing MoE-based GNNs on inductive and link prediction tasks.

## 1 Introduction

Graph-structured data emerge across many important domains—molecular chemistry, social networks, recommendation systems, biological interaction networks, and knowledge graphs—where relational structure and node/edge attributes jointly shape downstream prediction tasks. The rise of Graph Neural Networks (GNNs) has enabled powerful representation-learning on such data: by stacking message-passing layers, standard GNNs aggregate information across node neighborhoods and learn task-specific embeddings. However, despite the numerous successes, several key challenges remain when deploying GNNs in modern, large-scale, heterogeneous graph learning regimes. Early GNN research emphasised the *static* choice of the number of propagation layers. Subsequent work has shown that graph-scale heterogeneity, node degree and topological variability make the optimal propagation depth graph- and task-dependent. For example, decoupling receptive field size from layer depth mitigates oversmoothing and neighbourhood explosion in large graphs (Zeng et al., 2021; Gallicchio & Micheli, 2020; Poli et al., 2021). More recently, adaptive or continuous-depth GNNs (e.g., via graph differential equations) allow per-graph or per-node adjustment of propagation steps (Poli et al., 2021; Zheng et al., 2025). Yet, such dynamic-depth mechanisms have seen limited integration with parameter-efficient adaptation of pretrained graph models. Furthermore, the graph-learning ecosystem has shifted from ad-hoc per-graph GNN training toward graph foundation models (GFMs)—pretrained, large-scale graph models intended to support broad downstream adaptation across domains. For instance, GraphGPT presents large transformer-style models pretrained on graph data, demonstrating strong transfer potential with increasing importance of universal graph representations that generalise across structural heterogeneity and domain (Zhao et al.; Mao et al., 2024a;b). Equally, scalable graph-pretraining frameworks such as GPT-GNN show that

self-supervised generative pre-training on graphs enables improved downstream accuracy Hu et al. (2020c). Another complementary research direction is the rise of auto-GNN architectures and graph transformers: neural-architecture-search frameworks (GraphNAS, Auto-GNN) tailor GNN architectures to graph characteristics, relieving manual design effort Gao et al. (2019). Additionally, graph transformers extend the representational power of GNNs by integrating transformer blocks with graph structure, enabling larger receptive fields and structural flexibility. However, despite the progress, a research gap remains, in terms of, how to parameter-efficiently fine-tune GNN backbones in the face of graph-size, topology and domain heterogeneity, while also leveraging dynamic expert routing or depth-adaptive mechanisms without fully training large models from scratch.

We analyze the distribution of graph instances with respect to their order (number of nodes), as shown in Fig. 1 (a) and (b) for IMDB-BINARY and COLLAB, respectively. By computing the area under the curve, we partition the graphs into three equal groups such as small, medium, and large based on node counts, and create train–test splits for each. As shown in Fig. 1 (c), we observe a clear phenomenon of depth sensitivity. On small graphs, deeper GNNs often overfit, leading to redundant parameters and degraded performance. Conversely, shallow GNNs on large graphs under-reach, failing to capture global dependencies.

To counter depth sensitivity, recent work equips GNNs with Mixture-of-Experts (MoE) routers that dispatch each graph to depth or configuration specialized experts via a learnable gate, effectively aggregating across neighborhood radii and improving robustness under structural heterogeneity Yao et al. (2024). However, MoE incurs heavy parameter and compute overhead; end-to-end ensemble training is resource-intensive and vulnerable to overfitting and catastrophic forgetting, especially in low-label regimes Goodfellow et al. (2013). A pragmatic alternative is parameter-efficient fine-tuning (PEFT): freeze the backbone experts and insert lightweight adapters typically a down-projection, nonlinearity, and up-projection to

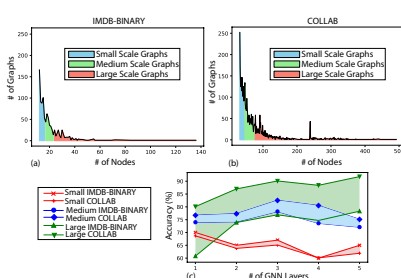

Figure 1: Comparison between the order of the graphs seen vs. the number of graphs in (a) IMDB-BINARY and (b) COLLAB. (c) The *depth-sensitivity* used in the IMDB-BINARY and COLLAB datasets shows the way different sized graphs depend on certain GNN depths for effective extraction of information.

capture task-specific shifts without perturbing pre-trained capacities Houlsby et al. (2019); He et al. (2022). PEFT preserves transferability, curbs forgetting, and cuts trainable parameters by over an order of magnitude, making MoE-style adaptability feasible in resource constrained settings.

In this paper, we propose an adapter-based Mixture-of-Experts model, MoLE-GNN (Mixture-of-Learnable Experts with Adapter GNN), which integrates parameter-efficient fine-tuning (PEFT) into GNNs - a challenging task - since most existing PEFT methods were originally developed for sequence models. MoLE-GNN demonstrates outstanding performance while tuning only a small fraction of parameters, even outperforming full fine-tuning. To enable this, we combine adapter modules with a dynamic gating mechanism, where adapters within each expert allow task-specific specialization while preserving generalization, and the gating network adaptively selects experts based on input graphs. This hybrid design balances efficiency and adaptability, delivering competitive performance under limited supervision while substantially reducing memory and compute costs. We demonstrate, combining adapters with dynamic MoE–GNNs yields a lightweight and robust framework that scales effectively across graph-, node-, and link-level tasks.

Our contributions are as follows - *i*) Dynamic-depth GNNs, GFMs, and auto-GNNs do not fully combine dynamic specialization and parameter-efficient fine-tuning in a unified framework. Our proposed MoLE-GNN integrates expert adapters with routing over heterogeneous graphs, enabling depth- and topology-aware fine-tuning with a small fraction of tunable parameters. *ii*) We empirically validate that MoLE-GNN outperforms baselines on diverse graph-scale tasks, achieving strong performance while lowering the trainable parameter count from 7.7M to just 0.39M, yet demonstrating robust transfer across graph-size heterogeneity.

## 2 RELATED WORK

For brevity, we focus here on recent works in mixture-of-experts models and graph-prompt tuning methods, while a comprehensive discussion is provided in Appendix A.

**Mixture of Experts Model.** The Mixture-of-Experts (MoE) framework Jacobs et al. (1991); Jordan & Jacobs (1994) trains specialized expert networks, with expert selection proposed via autoencoders Aljundi et al. (2017) and sparse gating Shazeer et al. (2017). MoE modules are now widely applied in vision Dai et al. (2021); Yu et al. (2024) and NLP Fedus et al. (2022); Du et al. (2022), and recently extended to GNNs. Examples include TopExpert Kim et al. (2023) (clustering-based gating), GMoE Wang et al. (2023) (multi-hop information), G-FAME Liu et al. (2023c) (fairness), Link-MoE Ma et al. (2024) and GraphMETRO Wu et al. (2023b) (task specialization and distribution shifts), and DA-MoE Yao et al. (2024) (adaptive depth for scale variation). However, existing MoE-GNNs are typically trained from scratch with large parameter counts rather than leveraging pre-trained GNN experts.

**Graph-Prompt Tuning Methods.** Prompt tuning methods, originating in NLP, adapt pre-trained models to downstream tasks by modifying inputs rather than model architecture Liu et al. (2021a); Lester et al. (2021). Variants include prefix-tuning Li & Liang (2021), which updates task-specific parameters per layer; adapter tuning Houlsby et al. (2019); Chen et al. (2022b), which inserts bottleneck adapters; BitFit Zaken et al. (2021), which tunes only bias terms; and LoRA Hu et al. (2022), which uses low-rank decomposition. These techniques have also been adopted in GNNs Wu et al. (2023c). Recently, AdapterGNN Li et al. (2024) extends adapter-based tuning to GNNs by integrating lightweight adapters into each layer, enabling efficient adaptation with minimal parameter updates. S2PGNN Zhili et al. (2024), fine-tunes both pre-trained backbone GNNs along with adapter and search best configuration of the architecture. Also GCNconv-Adapter Papageorgiou et al. (2025) present new graph adapter based model. However, existing parameter-efficient methods rely on a fixed layer configuration across all graphs, limiting their adaptability to varying data scales. To address this, we propose an adapter-based MoE framework that employs pre-trained GNNs as experts within a MoE architecture. By integrating tunable adapters into each expert, our approach mitigates challenges such as *depth sensitivity*, catastrophic forgetting, and overfitting, providing a more flexible and scalable fine-tuning strategy.

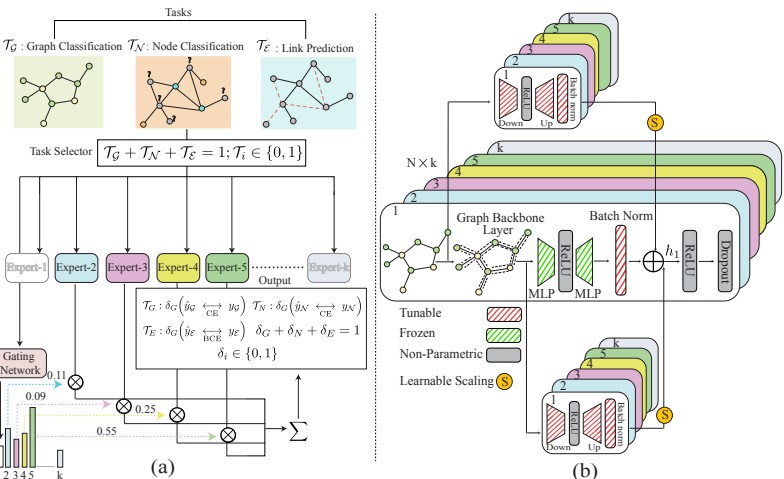

Figure 2: Overview of MoLE-GNN. (a) Multiple experts replace the conventional GNN backbone to model specialized patterns at various aggregation scales. Few of them are activated by the gating network, and the grey boxes indicate the inactive experts. (b) Each expert is a pre-trained graph encoder with its own layer configuration. Depending on the expert, this encoder may be a message-passing GNN or a Transformer-style attention architecture. We insert two parallel adapters that operate on the representations both before and after the core propagation/attention block, while the original encoder parameters remain frozen. This design enables each expert to combine its base architecture with lightweight, task-specific adapters and learnable scaling.

## 3 METHODOLOGY

MoLE-GNN, as depicted in Fig. 2, combines a dynamic MoE strategy with a PEFT approach through adapter modules. Fig. 2(a) shows that the model replaces a single, fixed GNN backbone with a collection of $K$ pre-trained experts, each specializing in different aggregation radii or depth configurations. For each input graph, the structure-aware gating network computes a sparse distribution over these experts and activates only a small subset, enabling the model to adapt its effective

receptive field to the graph's topology and scale. Fig. 2(b) shows the internal structure of each expert, where a frozen pre-trained GNN backbone is augmented with two lightweight adapter modules, one placed before and one after the message-passing or attention block to enable task-specific adaptation without altering backbone parameters. The two smaller groups of $K$ experts indicate that the gating network activates only a subset of experts for each input, while the remaining experts stay inactive. Our goal is to adapt powerful, pre-trained GNNs to diverse downstream tasks while updating only a small fraction of parameters. We consider three canonical regimes: (i) *graph classification* in the inductive setting, where entire graphs are mapped to labels; (ii) *node classification* in the transductive setting, where node labels are inferred on a fixed graph; and (iii) *link prediction*, where edge existence is predicted between node pairs. Let $G = (V, E)$, where $V$ is the set of nodes, and $E$ is the set of edges, be a graph with node features $X \in \mathbb{R}^{|V| \times D}$ and adjacency $A \in \{0, 1\}^{|V| \times |V|}$. We learn $f : \mathcal{G} \to \mathcal{Y}$ with $\mathcal{Y}$ defined by the downstream task, aiming for strong generalization across graph scales and topologies.

We instantiate $K$ frozen, pre-trained GNN experts $\{E_k(\cdot; \Theta_k)\}_{k=1}^{K}$, each specialized by depth or receptive field (e.g., shallow experts capture local neighborhoods while deeper experts model long-range dependencies). This heterogeneity directly addresses depth-sensitivity by allowing the system to favor shallow experts on small graphs and deeper experts on large graphs without retraining the full backbones. In all cases, $\Theta_k$ remain frozen during downstream adaptation.

**Adapter-based PEFT.** To envision relevant experts for a particular task while keeping $\Theta_k$ fixed, we introduce lightweight adapter modules. Given a hidden state $h \in \mathbb{R}^d$, the adapter is

$$A(h) \; = \; W_{\text{up}} \, \sigma\big(W_{\text{down}} \, h\big), \qquad W_{\text{down}} \in \mathbb{R}^{d \times r}, \; W_{\text{up}} \in \mathbb{R}^{r \times d}, \; r \ll d, \tag{1}$$

with nonlinearity $\sigma(\cdot)$ (e.g., ReLU). A learnable scalar $\alpha$ controls the adapter's residual contribution:

$$\tilde{h} \; = \; h \; + \; \alpha \, A(h). \tag{2}$$

To exploit structure explicitly and maintain numerical stability, we use the symmetrically normalized adjacency $\hat{A} = D^{-1/2}(A + I)D^{-1/2}$, where $D$, and $I$ are the degree and identity matrix, inside the adapter when operating on batched node/graph features $H \in \mathbb{R}^{|V| \times d}$:

$$A_{\text{graph}}(H) \; = \; W_{\text{up}} \, \sigma\big(\hat{A} \, H \, W_{\text{down}}\big), \qquad \tilde{H} \; = \; H \; + \; \alpha \, A_{\text{graph}}(H). \tag{3}$$

Adapters are placed (conceptually as in Fig. 2(b)) before and after message passing to specialize both feature transformation and neighborhood aggregation, without modifying the backbone.

**Dynamic expert routing.** A structure-aware gating network ($\Gamma(.)$) maps each input to a sparse combination of experts, as in Fig. 2(a). We compute a pooled graph representation $g = \text{READOUT}(G)$ and derive mixture weights via a small MLP and softmax:

$$w \; = \; \text{softmax}\big(\Gamma(g)\big) \in \mathbb{R}^K, \qquad \sum_{k=1}^{K} w_k = 1. \tag{4}$$

For computational efficiency and regularization, we retain only the top-$k_e$ entries of $w$, defining an active set $S$ of experts. Let $h_k$ denote expert $k$'s frozen output and $A_k$ its adapter; the aggregated representation is

$$z \; = \; \sum_{k \in S} w_k \left( h_k + \alpha_k \, A_k(h_k) \right), \tag{5}$$

which is then fed to a task head $f_{\text{head}}$.

**Task-specific heads and learning objectives.** We design distinct task-specific heads for inductive and transductive settings, as well as link prediction.

**Inductive Settings.** Node embeddings are aggregated with a pooling operator:

$$\hat{y} = \text{softmax}\big(W_g \cdot \text{POOL}(z)\big), \tag{6}$$

where $\text{POOL}(\cdot)$ can be mean, sum, or attention pooling.

**Transductive Settings.** Each node embedding $z_v$ is aligned with the fixed input topology, and classification is performed via:

$$\hat{y}_v = \text{softmax}(W_n z_v), \quad v \in V. \tag{7}$$

**Link Prediction.** For candidate pairs $(u, v)$, we use a bilinear decoder:

$$s(u, v) = z_u^\top W_\ell z_v. \tag{8}$$

Cross-entropy or ranking-based objectives (e.g., Hits@k, MRR) are applied accordingly.

**Theoretical guarantees: stability and graph-awareness.** To justify our design, we analyze two requirements: (i) *stability under perturbations*, ensuring controlled sensitivity, and (ii) *graph dependence*, ensuring that adapters leverage topology rather than acting as graph-agnostic bias terms.
**Scope.** We analyze stability only for the adapter-augmented *experts*. The routing (gating) module uses a standard GNN without adapters; during analysis we treat routing as fixed and make no claims about the gate.

**Theorem 3.1** (Stable Adapter-Augmented Experts). *Assume* $\|\hat{A}\|_2 \leq 1$, $\sigma$ *is* $L_\sigma$-*Lipschitz, and* $\|W_\downarrow\|_2 \leq \gamma_\downarrow$, $\|W_\uparrow\|_2 \leq \gamma_\uparrow$. *Then, for*

$$F(h) = h + \alpha W_\uparrow \sigma(\hat{A}hW_\downarrow),$$

*the Lipschitz constant satisfies*

$$Lip(F) \leq 1 + \alpha L_\sigma \gamma_\downarrow \gamma_\uparrow.$$

*Moreover, for MoE aggregation*

$$z = \sum_{k \in S} w_k F_k(h_k), \quad w \in \Delta_{K-1},$$

*we have*

$$Lip(z) \leq \max_{k \in S} \left( 1 + \alpha_k L_\sigma \gamma_\downarrow^{(k)} \gamma_\uparrow^{(k)} \right).$$

*Proof.* For a single adapter block,

$$\|W_\uparrow \sigma(\hat{A}W_\downarrow x) - W_\uparrow \sigma(\hat{A}W_\downarrow y)\| \leq \gamma_\uparrow L_\sigma \gamma_\downarrow \|x - y\|,$$

since $\|\hat{A}\|_2 \leq 1$. The residual connection adds an identity map, yielding Lipschitz constant $1 + \alpha L_\sigma \gamma_\downarrow \gamma_\uparrow$. For the MoE, $z$ is a convex (or sparse-convex) combination of Lipschitz maps. Such a combination inherits a Lipschitz constant bounded by the maximum of its components. $\square$

This theorem guarantees that adapters inject new flexibility without destabilizing training. By bounding the Lipschitz constant, we ensure robustness to small perturbations crucial in low-resource regimes where noisy or limited supervision may otherwise destabilize adaptation.

**Proposition 3.2** (Graph-Dependence of Adapters). *Let* $\hat{A}_1 \neq \hat{A}_2$ *be normalized adjacency matrices of two graphs* $G_1 \neq G_2$. *Consider*

$$A_{graph}(H) = W_\uparrow \sigma(\hat{A}HW_\downarrow).$$

*If* $W_\downarrow, W_\uparrow$ *are not rank-deficient, then for a non-measure-zero set of* $H$,

$$A_{graph}^{(1)}(H) \neq A_{graph}^{(2)}(H).$$

*Proof.* If $\hat{A}_1 \neq \hat{A}_2$, then $(\hat{A}_1 - \hat{A}_2)HW_\downarrow \neq 0$ for some $H$. Since $\sigma$ is piecewise-linear and $W_\uparrow$ is non-degenerate, the images differ on an open set of $H$, yielding non-trivial dependence on $G$. $\square$

This proposition establishes that our adapters genuinely exploit graph structure. Unlike graph agnostic adapters, which produce identical transformations regardless of topology, our design ensures responses vary with adjacency, enabling task-specific specialization across diverse graph topologies. Together, Theorem 3.1 and Proposition 3.2 formalize why MoLE-GNN is both *stable* (avoiding uncontrolled growth or sensitivity) and *structurally adaptive*, providing the theoretical underpinnings for its effectiveness in inductive, transductive, and link-level tasks.

# 4 EXPERIMENTAL SETUP

**Dataset.** We construct the pre-training corpus from two million unlabeled molecules in ZINC15 Sterling & Irwin (2015), 395K protein ego-networks from PPI, and academic and social graphs from NetRep and SNAP Ritchie et al. (2016); Leskovec & Sosič (2016). Each expert GNN is pre-trained on this corpus, following prior work Hu et al. (2020b); Qiu et al. (2020). We adopt the ogbn-arxiv dataset with GraphMAE Hou et al. (2022) to pre-train expert GNNs for node classification and link prediction. Downstream evaluation covers 14 datasets for inductive setting, 5 for transductive setting, and 4 for link prediction. Detailed dataset statistics are provided in Tables 9, 10 (Appendix B).

**Evaluation Metrics.** Here, we follow the work by Li et al. (2024) and use ROC-AUC as the evaluation metrics for inductive learning experiments. For transductive learning experiment, we utilize work Accuracy as the evaluation metric Papageorgiou et al. (2025). For, link prediction we use MRR and Hits@20 as the evaluation metric Ma et al. (2024).

**Baseline Frameworks.** For inductive learning experiments, we evaluate full fine-tuning, two graph prompt learning methods, four widely used PEFT models, and a mixture-of-experts approach: GPF Fang et al. (2022), MolCPT Diao et al. (2022), Adapter Chen et al. (2022b), LoRA, AdapterGNN, GCNConv-Adapter, and TopExpert. Additionally, we consider Surgical Fine-Tuning Lee et al. (2022) and BitFit for transductive learning. For link prediction, we use all of the above baselines and further include the MoE-based Link-MoE. For all these three type of tasks, we conisder GMoE, and DA-MoE as the general purpose graph MoE based methods as baseline models as described in Appendix C.

| Pre-training Method | Tuning Method | BACE | BBBP | ClinTox | HIV | SIDER | Tox21 | MUV | ToxCast | PPI | Avg. |
|---|---|---|---|---|---|---|---|---|---|---|---|
| | | **Datasets (ROC-AUC ↑)** | | | | | | | | | |
| EdgePred | Full Fine-tune (100%) | 79.9±0.9 | 67.3±2.4 | 64.1±3.7 | 76.3±1.0 | 60.4±0.7 | **76.0**±0.6 | 74.1±2.1 | 64.1±0.6 | 65.6±0.9 | 69.8 |
| | Adapter (5.2%) | 78.5±1.7 | 65.9±2.8 | 66.6±5.4 | 73.5±0.2 | 60.9±1.3 | 75.4±0.5 | 73.0±1.0 | 63.0±0.7 | 69.8±0.5 | 69.6 |
| | LoRA (5.0%) | **81.0**±0.8 | 64.8±1.6 | _67.7_±1.2 | 74.8±1.2 | 60.8±1.1 | 74.6±0.4 | 75.0±1.5 | 62.2±1.0 | 68.0±1.0 | 69.9 |
| | GPF (0.1%) | 68.0±0.4 | 55.9±0.2 | 50.8±0.1 | 66.0±0.7 | 51.5±0.7 | 63.1±0.5 | 63.1±0.1 | 55.7±0.5 | 51.2±1.3 | 58.3 |
| | AdapterGNN (5.2%) | 79.0±1.5 | 69.7±1.4 | _67.7_±3.0 | _76.4_±0.7 | 61.2±0.9 | 75.9±0.9 | 75.8±2.1 | _64.2_±0.5 | _70.6_±1.1 | 71.2 |
| | TopExpert (100%) | _80.2_±0.8 | 66.2±0.8 | 56.8±2.4 | 76.0±0.7 | 59.6±0.5 | 74.1±0.4 | **79.9**±1.2 | 62.5±0.4 | 66.3±0.9 | 69.1 |
| | GCNconv-Adapter (3.0%) | 75.9±3.4 | _70.0_±1.2 | 51.9±2.4 | 68.6±4.6 | 60.1±1.4 | 72.7±0.5 | 66.7±2.7 | 61.8±1.1 | 70.3±1.9 | 66.4 |
| | **MoLE-GNN (ours) (5.1%)** | **81.0**±0.7 | **73.9**±0.5 | **75.5**±1.4 | **77.6**±0.9 | **62.5**±0.8 | 75.8±0.4 | _78.5_±1.1 | **64.9**±0.3 | **72.7**±0.6 | **73.6** |
| ContextPred | Full Fine-tune (100%) | 79.6±1.2 | 68.0±2.0 | 65.9±3.8 | _77.3_±1.0 | 60.9±0.6 | **75.7**±0.7 | 75.8±1.7 | _63.9_±0.6 | 63.5±1.1 | 70.1 |
| | Adapter (5.2%) | 75.0±3.3 | 68.2±3.0 | 57.6±3.6 | 75.4±0.6 | **62.4**±1.2 | 74.7±0.7 | 73.3±0.8 | 62.2±0.4 | 68.2±1.5 | 68.6 |
| | LoRA (5.0%) | 78.5±1.1 | 65.3±2.4 | 61.3±1.9 | 74.7±1.6 | 60.8±0.4 | 72.9±0.4 | 75.4±0.9 | 63.4±0.2 | 68.0±1.1 | 68.9 |
| | GPF (0.1%) | 58.7±0.6 | 58.6±0.6 | 39.8±0.8 | 68.0±0.4 | 59.4±0.2 | 67.8±0.9 | 71.8±0.8 | 58.8±0.5 | 67.1±0.6 | 61.1 |
| | AdapterGNN (5.2%) | 78.7±2.0 | 68.2±2.9 | _68.7_±5.3 | 76.1±0.5 | 61.1±1.0 | _75.4_±0.6 | 76.3±1.0 | 63.2±0.3 | _68.3_±1.5 | 70.7 |
| | TopExpert (100%) | _80.4_±1.4 | 69.9±0.8 | 58.9±4.3 | **78.2**±0.3 | 60.2±0.6 | 73.9±0.3 | **79.9**±0.9 | 62.9±0.3 | 56.3±1.0 | 69.0 |
| | GCNconv-Adapter (3.0%) | 79.8±2.0 | _70.1_±0.6 | 53.0±5.2 | 73.9±1.2 | 59.9±1.1 | 72.4±0.7 | 72.6±2.2 | 61.2±0.9 | 67.3±1.6 | 67.8 |
| | **MoLE-GNN (ours) (5.1%)** | **80.8**±0.5 | **73.1**±0.4 | **79.8**±0.7 | 77.3±0.8 | _62.4_±0.7 | 75.1±0.4 | _79.2_±1.0 | **64.1**±0.3 | **70.9**±0.7 | **73.6** |
| AttrMasking | Full Fine-tune (100%) | 79.3±1.6 | 64.3±2.8 | 71.8±4.1 | _77.2_±1.1 | 61.0±0.7 | **76.7**±0.4 | 74.7±1.4 | _64.2_±0.6 | 63.2±1.2 | 70.3 |
| | Adapter (5.2%) | 76.1±1.4 | 68.7±1.7 | 65.8±4.4 | 75.6±0.7 | 59.8±1.7 | 74.4±0.9 | 75.8±2.4 | 62.6±0.8 | _70.9_±1.0 | 70.0 |
| | LoRA (5.0%) | 79.8±0.7 | 64.2±1.1 | 70.1±2.9 | 76.1±1.4 | 59.7±0.5 | 74.6±0.5 | 76.6±1.6 | 61.7±0.4 | 69.2±0.8 | 70.2 |
| | GPF (0.1%) | 61.7±0.3 | 54.3±0.3 | 56.4±0.2 | 64.0±0.2 | 52.0±0.2 | 69.2±0.3 | 62.9±0.9 | 58.1±1.0 | 60.8±0.3 | 60.8 |
| | AdapterGNN (5.2%) | 79.7±1.3 | 67.5±2.2 | _78.3_±2.6 | 76.7±1.2 | _61.3_±1.1 | 76.6±0.5 | **78.4**±0.7 | 63.6±0.5 | 69.7±1.1 | 72.4 |
| | TopExpert (100%) | _81.3_±1.2 | _71.4_±0.7 | 70.1±1.3 | 77.1±0.7 | 60.3±0.6 | 75.5±0.3 | _78.4_±1.4 | 62.8±0.2 | 60.1±1.3 | 70.8 |
| | GCNconv-Adapter (3.0%) | 78.4±3.7 | 71.3±1.4 | 51.1±5.0 | 71.9±0.9 | 59.2±1.3 | 72.6±0.7 | 70.0±1.8 | 61.9±1.1 | 68.1±1.9 | 67.2 |
| | **MoLE-GNN (ours) (5.1%)** | **81.6**±0.8 | **73.2**±0.8 | **80.0**±1.5 | **78.2**±0.6 | **62.8**±0.6 | 76.1±0.3 | **79.2**±1.2 | **64.3**±0.2 | **71.2**±1.0 | **74.1** |
| GraphCL | Full Fine-tune (100%) | 74.6±2.2 | 68.6±2.3 | 69.8±7.2 | 78.5±1.2 | 59.6±0.7 | 74.4±0.5 | 73.7±2.7 | 62.9±0.4 | 65.5±0.8 | 69.7 |
| | Adapter (5.2%) | 72.5±3.0 | 69.3±0.6 | 67.3±7.4 | 75.0±0.4 | 59.7±1.2 | 74.7±0.4 | 72.9±1.7 | 62.9±0.4 | 69.0±0.8 | 69.3 |
| | LoRA (5.0%) | 75.1±0.7 | 67.8±1.1 | 65.1±3.5 | _78.9_±0.6 | 57.6±0.7 | 73.9±0.9 | 72.8±1.2 | 62.7±0.6 | _69.4_±0.6 | 69.3 |
| | GPF (0.1%) | 71.5±0.6 | 63.7±0.4 | 64.5±0.6 | 70.3±0.5 | 55.3±0.6 | 65.5±0.5 | 70.1±0.7 | 58.5±0.5 | 62.3±0.5 | 64.6 |
| | AdapterGNN (5.2%) | 76.1±2.2 | 67.8±1.4 | **72.0**±3.8 | 77.8±1.3 | 59.6±1.3 | _74.9_±0.9 | 75.1±2.1 | 63.1±0.4 | 68.1±1.5 | 70.5 |
| | TopExpert (100%) | _77.9_±1.3 | 70.9±0.8 | _70.7_±3.8 | **80.7**±0.7 | 60.1±1.0 | 74.6±0.5 | **78.2**±1.2 | 62.3±0.5 | 62.3±1.1 | _70.9_ |
| | GCNconv-Adapter (3.0%) | **79.8**±2.5 | _71.8_±0.8 | 54.3±3.3 | 72.2±1.5 | 59.8±1.4 | 72.5±0.7 | 75.2±0.1 | 62.0±0.8 | 66.3±0.6 | 68.2 |
| | **MoLE-GNN (ours) (5.1%)** | _79.8_±1.3 | **72.0**±0.8 | 69.8±5.3 | 76.8±0.8 | **61.9**±0.6 | **75.3**±0.4 | _76.4_±1.1 | **64.3**±0.4 | **71.1**±1.5 | **71.9** |
| SimGRACE | Full Fine-tune (100%) | 74.7±1.0 | 69.0±1.0 | 59.9±2.3 | 74.6±1.2 | 59.1±0.6 | 73.9±0.4 | 71.0±1.9 | 61.8±0.4 | 68.2±1.2 | 68.0 |
| | Adapter (5.2%) | 73.4±1.1 | 64.8±0.7 | 63.5±4.4 | 73.9±1.0 | _59.9_±0.9 | 73.1±0.9 | 70.1±1.4 | 61.7±0.8 | 64.5±2.0 | 67.2 |
| | LoRA (5.0%) | 73.2±1.0 | 67.5±0.4 | 60.7±0.4 | 74.1±0.5 | 57.6±2.6 | 72.2±0.2 | 67.9±0.9 | 61.8±0.2 | 63.0±0.3 | 66.5 |
| | AdapterGNN (5.2%) | _77.7_±1.7 | 68.1±1.3 | _73.9_±7.0 | 75.1±1.2 | 58.9±0.9 | **74.4**±0.6 | 71.8±1.4 | 62.6±0.6 | _70.1_±1.2 | _70.3_ |
| | TopExpert (100%) | 74.0±1.0 | 65.3±1.2 | 56.9±2.5 | 73.6±2.5 | 56.3±0.7 | 71.5±0.3 | _73.6_±1.1 | 61.9±0.1 | 65.8±1.4 | 66.5 |
| | GCNconv-Adapter (3.0%) | 77.4±2.0 | _70.5_±1.7 | 50.9±3.3 | _76.6_±0.4 | 59.8±1.4 | 72.8±0.8 | 68.5±4.0 | 61.9±1.0 | 64.3±1.8 | 67.0 |
| | **MoLE-GNN (ours) (5.1%)** | **81.7**±0.9 | **71.9**±0.7 | **78.8**±1.7 | **77.4**±0.4 | **60.8**±0.4 | _75.2_±0.3 | **75.8**±1.2 | **63.9**±0.5 | **70.7**±1.2 | **72.9** |

Table 1: Test ROC-AUC (%) performance on molecular and PPI prediction benchmarks using different tuning methods and pre-trained GNN models. Results are reported as mean ± standard deviation of ROC-AUC. Best performing model, based on average ROC-AUC (%) is shown in **bold**, while the second-best model is underlined

# 5 RESULTS & ANALYSIS

In the following section, we empirically validate our framework and present fine-grained results in Table 1, Table 5, Table 6, and Figure 3. Implementation details are provided in Appendix E, and the choice of pre-trained models is discussed in Appendix D. The base configuration of MoLE-GNN employs five experts for inductive learning, three experts for transductive learning, and link

| Pre-training Method | Tuning Method | Datasets (ROC-AUC ↑) | | | | | |
|---|---|---|---|---|---|---|---|
| | | BACE | BBBP | ClinTox | SIDER | Tox21 | ToxCast |
| EdgePred | S2PGNN (100%) | $82.2_{\pm1.1}$ | $69.1_{\pm0.8}$ | $71.9_{\pm1.1}$ | $62.3_{\pm0.5}$ | $77.1_{\pm0.8}$ | $66.2_{\pm0.3}$ |
| | S2PGNN (5.2%) | $80.7_{\pm1.1}$ | $65.1_{\pm0.8}$ | $75.1_{\pm2.2}$ | $57.1_{\pm0.8}$ | $75.1_{\pm0.7}$ | $61.2_{\pm0.5}$ |
| | **MoLE-GNN (ours) (5.1%)** | $81.0_{\pm0.7}$ | $73.9_{\pm0.5}$ | $75.5_{\pm1.4}$ | $62.5_{\pm0.8}$ | $75.8_{\pm0.4}$ | $64.9_{\pm0.3}$ |
| ContextPred | S2PGNN (100%) | $82.6_{\pm0.7}$ | $70.9_{\pm1.3}$ | $75.9_{\pm2.2}$ | $62.8_{\pm0.3}$ | $76.3_{\pm0.4}$ | $67.0_{\pm0.5}$ |
| | S2PGNN (5.2%) | $75.4_{\pm1.1}$ | $65.4_{\pm0.5}$ | $75.3_{\pm2.2}$ | $61.8_{\pm0.4}$ | $73.3_{\pm0.4}$ | $62.7_{\pm0.1}$ |
| | **MoLE-GNN (ours) (5.1%)** | $80.8_{\pm0.5}$ | $73.1_{\pm0.4}$ | $79.8_{\pm0.7}$ | $62.4_{\pm0.7}$ | $75.1_{\pm0.4}$ | $64.1_{\pm0.3}$ |
| AttrMasking | S2PGNN (100%) | $82.7_{\pm0.8}$ | $71.9_{\pm1.1}$ | $74.8_{\pm3.1}$ | $62.9_{\pm0.4}$ | $77.3_{\pm0.4}$ | $66.8_{\pm0.5}$ |
| | S2PGNN (5.2%) | $77.2_{\pm0.8}$ | $70.4_{\pm1.1}$ | $72.8_{\pm3.1}$ | $61.2_{\pm0.4}$ | $75.3_{\pm0.2}$ | $61.0_{\pm0.1}$ |
| | **MoLE-GNN (ours) (5.1%)** | $81.6_{\pm0.8}$ | $73.2_{\pm0.8}$ | $80.0_{\pm1.5}$ | $62.8_{\pm0.6}$ | $76.1_{\pm0.3}$ | $64.3_{\pm0.2}$ |
| GraphCL | S2PGNN (100%) | $82.6_{\pm2.3}$ | $70.8_{\pm1.1}$ | $75.2_{\pm3.3}$ | $62.4_{\pm1.2}$ | $76.8_{\pm0.5}$ | $66.6_{\pm0.3}$ |
| | S2PGNN (5.2%) | $77.1_{\pm2.3}$ | $68.4_{\pm1.1}$ | $68.9_{\pm3.4}$ | $59.4_{\pm1.2}$ | $73.2_{\pm0.5}$ | $63.5_{\pm0.3}$ |
| | **MoLE-GNN (ours) (5.1%)** | $79.8_{\pm1.3}$ | $72.0_{\pm0.8}$ | $69.8_{\pm5.3}$ | $61.9_{\pm0.6}$ | $75.3_{\pm0.4}$ | $64.3_{\pm0.4}$ |
| SimGRACE | S2PGNN (100%) | $83.9_{\pm1.5}$ | $69.3_{\pm0.9}$ | $73.6_{\pm3.3}$ | $62.3_{\pm0.6}$ | $75.9_{\pm0.2}$ | $65.8_{\pm0.5}$ |
| | S2PGNN (5.2%) | $80.2_{\pm1.5}$ | $69.3_{\pm0.9}$ | $73.6_{\pm3.3}$ | $55.7_{\pm0.6}$ | $74.8_{\pm0.2}$ | $61.4_{\pm0.1}$ |
| | **MoLE-GNN (ours) (5.1%)** | $81.7_{\pm0.9}$ | $71.9_{\pm0.7}$ | $78.8_{\pm1.7}$ | $60.8_{\pm0.4}$ | $75.2_{\pm0.3}$ | $63.9_{\pm0.5}$ |

Table 2: MoLE-GNN versus S2PGNN: performance comparison between MoLE-GNN and S2PGNN's adapter-based fine-tuning search. The best performing model is **bold** and second best performing model is underlined.

| Pre-training Method | Tuning Method | ogbn-arxiv (Accuracy) | ogbn-proteins (ROC-AUC) | ogbn-products (Accuracy) |
|---|---|---|---|---|
| NodeFormer | Full Fine-tune (100%) | $58.5_{\pm0.2}$ | $77.5_{\pm1.2}$ | $62.6_{\pm0.1}$ |
| | AdapterGNN (6.8%) | $64.9_{\pm0.4}$ | $75.1_{\pm0.5}$ | $65.3_{\pm0.4}$ |
| | GCNconv-Adapter (3.0%) | $56.2_{\pm0.7}$ | $71.5_{\pm0.6}$ | $27.2_{\pm0.3}$ |
| | **MoLE-GNN (ours) (3.6%)** | $67.5_{\pm0.2}$ | $77.6_{\pm0.3}$ | $67.5_{\pm0.2}$ |
| DIFFormer-s | Full Fine-tune (100%) | $47.8_{\pm0.9}$ | $72.5_{\pm0.4}$ | $55.2_{\pm0.4}$ |
| | AdapterGNN (6.8%) | $54.9_{\pm1.3}$ | $58.4_{\pm0.4}$ | $54.9_{\pm0.3}$ |
| | GCNconv-Adapter (3.0%) | $26.7_{\pm0.9}$ | $65.1_{\pm1.1}$ | $22.9_{\pm0.9}$ |
| | **MoLE-GNN (ours) (3.6%)** | $58.2_{\pm0.6}$ | $75.2_{\pm0.4}$ | $65.2_{\pm0.9}$ |

Table 3: Performance comparison on large-scale node classification benchmarks across different fine-tuning methods on pretrained graph transformers. Best scores are in bold, and second-best are underlined.

| Tuning Methods | Methods | Datasets (Time in Seconds) | | | | | | | # Trainable Params(M) |
|---|---|---|---|---|---|---|---|---|---|
| | | BBBP | Tox21 | ToxCast | SIDER | ClinTox | BACE | Avg. | |
| Full Fine-Tune | GIN | 2.31 | 2.88 | 2.34 | 1.98 | 1.61 | 2.21 | 2.22 | 7.81 |
| | GCN | 1.75 | 2.55 | 2.61 | 1.68 | 1.88 | 2.03 | 2.08 | 2.65 |
| | GAT | 2.23 | 2.74 | 3.39 | 1.97 | 2.02 | 2.36 | 2.45 | 4.45 |
| | **MoLE-GNN** | 0.45 | 0.62 | 0.58 | 0.45 | 0.53 | 0.47 | 0.52 | 0.39 |
| Adapter Tuning | Adapter-GNN | 1.67 | 1.42 | 1.49 | 0.52 | 0.51 | 0.52 | 1.02 | 0.12 |
| | GCNConv-Adapter | 0.64 | 1.34 | 1.65 | 0.49 | 0.48 | 0.58 | 0.86 | 0.05 |
| | S2PGNN | 6.65 | 8.25 | 9.15 | 6.65 | 9.00 | 7.00 | 7.78 | 8.12 |
| | **MoLE-GNN** | 0.45 | 0.62 | 0.58 | 0.45 | 0.53 | 0.47 | 0.52 | 0.39 |
| MoE Tuning | GMoE | 2.03 | 3.12 | 3.70 | 1.90 | 1.22 | 1.02 | 2.17 | 14.9 |
| | DA-MoE | 1.63 | 2.26 | 3.02 | 1.27 | 0.89 | 1.03 | 1.68 | 29.8 |
| | TopExpert | 1.19 | 2.02 | 2.36 | 0.56 | 0.62 | 0.51 | 1.21 | 2.51 |
| | **MoLE-GNN** | 0.45 | 0.62 | 0.58 | 0.45 | 0.53 | 0.47 | 0.52 | 0.39 |

Table 4: Detailed time and memory comparison of MoLE-GNN and other baseline methods on graph classification dataset. Here, M stands for million. The lowest time and parameter count are marked in **bold**, and the second-lowest are underlined.

| Pre-training Method | Tuning Method | Datasets (Accuracy ↑) | | | | | |
|---|---|---|---|---|---|---|---|
| | | Cora | Citeseer | PubMed | Wisconsin | Texas | Avg. |
| NodeFormer | Full Fine-tune (100%) | $85.1_{\pm0.8}$ | $77.0_{\pm1.7}$ | $87.9_{\pm0.2}$ | $59.1_{\pm3.8}$ | $60.9_{\pm8.7}$ | $74.0$ |
| | Surgical Fine Tuning (0.15%) | $78.3_{\pm1.3}$ | $75.5_{\pm2.2}$ | $87.8_{\pm0.4}$ | $53.1_{\pm4.9}$ | $60.4_{\pm4.8}$ | $71.0$ |
| | BitFit (0.10%) | $78.2_{\pm0.9}$ | $74.6_{\pm2.4}$ | $87.9_{\pm0.6}$ | $54.4_{\pm7.9}$ | $58.7_{\pm8.7}$ | $70.8$ |
| | LoRA (1.42%) | $85.3_{\pm0.8}$ | $76.2_{\pm1.4}$ | $87.4_{\pm0.1}$ | $53.4_{\pm3.2}$ | $62.6_{\pm7.6}$ | $72.9$ |
| | Adapter (1.00%) | $78.5_{\pm1.3}$ | $74.5_{\pm2.6}$ | $87.9_{\pm0.5}$ | $56.6_{\pm4.9}$ | $63.8_{\pm6.7}$ | $72.3$ |
| | G-Adapter (1.20%) | $83.7_{\pm0.7}$ | $74.3_{\pm2.3}$ | $89.0_{\pm0.3}$ | $52.8_{\pm3.6}$ | $62.6_{\pm5.3}$ | $72.5$ |
| | AdapterGNN (6.8%) | $78.7_{\pm1.2}$ | $74.9_{\pm2.1}$ | $88.3_{\pm0.5}$ | $60.9_{\pm8.3}$ | $63.4_{\pm7.2}$ | $73.3$ |
| | GCNconv-Adapter (3.0%) | $81.3_{\pm0.9}$ | $76.6_{\pm2.0}$ | $89.1_{\pm0.1}$ | $55.3_{\pm5.0}$ | $64.4_{\pm4.3}$ | $73.3$ |
| | **MoLE-GNN (ours) (3.6%)** | $85.6_{\pm1.0}$ | $77.3_{\pm0.7}$ | $89.3_{\pm0.4}$ | $72.5_{\pm8.5}$ | $72.3_{\pm7.3}$ | $79.4$ |
| DIFFormer-s | Full Fine-tune (100%) | $83.2_{\pm5.0}$ | $73.2_{\pm1.6}$ | $87.9_{\pm0.5}$ | $48.1_{\pm6.5}$ | $58.7_{\pm3.7}$ | $70.2$ |
| | Surgical Fine Tuning (0.15%) | $81.8_{\pm5.3}$ | $73.5_{\pm1.2}$ | $87.3_{\pm3.8}$ | $50.6_{\pm6.1}$ | $57.5_{\pm4.7}$ | $70.1$ |
| | BitFit (0.10%) | $81.4_{\pm5.5}$ | $48.8_{\pm6.1}$ | $76.0_{\pm6.1}$ | $49.1_{\pm8.5}$ | $60.0_{\pm5.3}$ | $63.1$ |
| | LoRA (1.42%) | $78.9_{\pm6.2}$ | $70.5_{\pm1.1}$ | $84.7_{\pm4.3}$ | $45.6_{\pm8.2}$ | $57.5_{\pm4.5}$ | $67.4$ |
| | Adapter (1.00%) | $81.4_{\pm5.3}$ | $72.9_{\pm1.8}$ | $85.2_{\pm3.9}$ | $47.2_{\pm5.9}$ | $55.8_{\pm5.8}$ | $68.5$ |
| | G-Adapter (1.20%) | $67.9_{\pm17.5}$ | $74.1_{\pm1.4}$ | $73.3_{\pm15.4}$ | $57.2_{\pm5.8}$ | $62.1_{\pm1.6}$ | $66.9$ |
| | AdapterGNN (6.8%) | $80.1_{\pm6.7}$ | $70.1_{\pm1.4}$ | $86.8_{\pm0.6}$ | $69.7_{\pm5.9}$ | $56.2_{\pm2.6}$ | $72.6$ |
| | GCNconv-Adapter (3.0%) | $82.8_{\pm5.0}$ | $72.1_{\pm1.9}$ | $85.7_{\pm0.5}$ | $57.8_{\pm4.7}$ | $62.9_{\pm5.9}$ | $72.3$ |
| | **MoLE-GNN (ours) (3.6%)** | $83.5_{\pm0.9}$ | $73.6_{\pm1.4}$ | $88.2_{\pm0.3}$ | $76.3_{\pm3.7}$ | $71.9_{\pm5.3}$ | $78.7$ |

Table 5: Test Accuracy (%) performances on node classification benchmarks with different tuning methods and pre-trained graph transformers models. Best performing model, based on average Accuracy (%) is shown in **bold**, while the second-best model is underlined

| Methods | Datasets | | | |
|---|---|---|---|---|
| | Cora MRR ↑ | Citeseer MRR ↑ | PubMed MRR ↑ | ogbl-ddi Hits@20 ↑ |
| Full Fine-tune (100%) | $8.9_{\pm7.04}$ | $13.1_{\pm3.4}$ | $7.3_{\pm3.1}$ | $83.3_{\pm3.3}$ |
| Surgical Fine Tuning (0.15%) | $24.3_{\pm14.4}$ | $26.4_{\pm2.5}$ | $19.3_{\pm2.7}$ | $24.2_{\pm14.5}$ |
| BitFit (0.20%) | $19.6_{\pm9.5}$ | $29.5_{\pm6.5}$ | $10.3_{\pm3.1}$ | $19.5_{\pm17.2}$ |
| LoRA (7.0%) | $11.1_{\pm3.1}$ | $16.7_{\pm3.4}$ | $3.8_{\pm1.1}$ | $16.5_{\pm4.0}$ |
| Adapter (7.0%) | $26.0_{\pm8.5}$ | $39.2_{\pm12.6}$ | $20.5_{\pm9.1}$ | $40.9_{\pm18.1}$ |
| G-Adapter (7.8%) | $28.0_{\pm5.9}$ | $33.2_{\pm9.7}$ | $21.6_{\pm7.1}$ | $38.1_{\pm1.3}$ |
| AdapterGNN (1.7%) | $25.1_{\pm6.2}$ | $28.8_{\pm5.8}$ | $15.8_{\pm4.6}$ | $72.5_{\pm6.3}$ |
| GCNconv-Adapter (1.5%) | $28.6_{\pm7.4}$ | $31.2_{\pm6.1}$ | $29.9_{\pm4.6}$ | $78.4_{\pm3.7}$ |
| Link-MoE (100%) | $44.0_{\pm2.3}$ | $64.6_{\pm3.7}$ | $53.1_{\pm0.2}$ | $85.2_{\pm1.3}$ |
| **MoLE-GNN (2.3%)** | $49.3_{\pm4.0}$ | $66.4_{\pm4.6}$ | $44.7_{\pm4.2}$ | $93.2_{\pm1.3}$ |

Table 6: Test performance on the link prediction benchmark with different tuning methods. The best perform model, based MRR and Hits@20 (mean ± standard deviation) are shown in **bold**, while the second-best model is underlined.

prediction. Each expert is instantiated as either a Graph Isomorphism Network (GIN) or a Graph Transformers (GTs) equipped with sequential adapters (placed before and after the message passing).

**Inductive Learning Results.** We evaluate MoLE-GNN under inductive learning settings, as reported in Table 1. Across all datasets, MoLE-GNN consistently outperforms full fine-tuning in classification. MoLE-GNN achieves higher ROC–AUC than full fine-tuning across ClinTox, SIDER, BACE, BBBP, HIV, ToxCast, and PPI, yielding an overall average of 73.22% i.e, 5.2% average improvement over 69.6%. Furthermore, MoLE-GNN outperforms the domain-specific TopExpert on eight of nine datasets and achieves comparable performance on MUV, yielding an average ROC–AUC of 73.2% a 5.8% improvement over TopExpert (69.2%). Furthermore, MoLE-GNN surpasses existing state-of-the-art (SOTA) graph-specific PEFT methods outperforming GCNconv-Adapter by 8.7% and AdapterGNN by 3.1% and delivers notable improvements over conventional PEFT approaches. A main reason MoLE-GNN surpasses pre-trained PEFT approaches, and TopExpert is its structure-aware sparse routing to depth specific experts with lightweight, topology-conditioned adapters, aligning receptive fields to graph scale and reducing overfitting for stronger inductive generalization. For brevity, we report the additional results of MoLE-GNN on social

network graph classification (Table 11) and few-shot learning (Fig. 5) in Appendix F. We also evaluate MoLE-GNN against the automatic adapter-based fine-tuning framework S2PGNN Zhili et al. (2024), and the results are presented in Table 2. From Table 2, we observe that MoLE-GNN consistently outperforms S2PGNN on BBBP across all pre-training methods, achieves better performance on ClinTox for four out of five pre-training methods, and produces comparable results under the GraphCL pre-training setting. For the remaining datasets, MoLE-GNN produces comparable results. A key reason for the comparable performance is that S2PGNN fine-tunes the entire pre-trained backbone together with its identity-augmentation modules including the bottleneck blocks that resemble adapter tuning and additionally searches for the best fine-tuning configuration for each dataset. In contrast, MoLE-GNN freezes the backbone of each expert and tunes only lightweight adapters within a fixed architecture. Despite this more restricted tuning regime and the absence of any search, MoLE-GNN still surpasses S2PGNN on BBBP and ClinTox. Moreover, from Table 4, we observe that the S2PGNN search procedure requires substantially more training time and 20× more tunable parameters. For a fair comparison, we evaluate a controlled variant of S2PGNN in which all backbone parameters of the pre-trained GNNs are frozen and only the adapter-like modules are tuned. Under this setting, we observe that MoLE-GNN outperforms S2PGNN across all datasets for all pre-training methods.

**Transductive Learning Results.** We evaluate the performance of MoLE-GNN under transductive learning, with results summarized in Table 5. Across the three homophilic benchmarks Cora, Citeseer, and PubMed MoLE-GNN consistently outperforms fully fine-tuned baselines (NodeFormer and DIFFormer-s), providing an average gain of 0.7%. Again, we observe that on the two heterophilic datasets, Wisconsin and Texas, our MoLE-GNN, instantiated with either pre-trained Node-Former or DIFFormer-s (pre-trained on the homophilic dataset ogbn-arxiv), consistently outperforms vanilla fine-tuning. This demonstrates that each expert in MoLE-GNN transfers pre-trained knowledge more effectively than the vanilla fine-tuning approach. Overall, this achieves an average Accuracy of 79.1%, outperforming vanilla fine-tuning, where both NodeFormer and DIFFormer-s attain only 72.1%. Furthermore, MoLE-GNN outperforms SOTA graph specific PEFT methods, GCNconv-Adapter by 8.58% and AdapterGNN by 8.36%, and produces substantial gains over classical PEFT techniques. A primary reason is MoLE-GNN excels its structure aware gating, which routes inputs to adapter based graph transformers within each expert, capturing richer graph structure than SOTA baselines. In addition, we provide a comparison of MoLE-GNN with existing graph prompt based baselines (Table 12) and report its few-shot performance (Fig. 5) in Appendix F.

**Transductive Learning results for Large-Scale Graphs.** We examine the scalability of MoLE-GNN on large-scale transductive node classification tasks such as protein-protein interaction, citation network and product co-purchase tasks using three large benchmark datasets: ogbn-arxiv, ogbn-proteins, and ogbn-products. The results are summarized in Table 3. As shown in the table, MoLE-GNN consistently outperforms all full fine-tuning baselines. Notably, when applied on top of the NodeFormer pretraining backbone, MoLE-GNN achieves a 0.14% ROC-AUC improvement on ogbn-proteins, despite updating only a small fraction of parameters. This demonstrates that our parameter-efficient updates can surpass full-model fine-tuning even on extremely large graphs. For the remaining two datasets, MoLE-GNN also outperforms the full fine-tuning setting in terms of Accuracy, achieving improvements of 15.97% on ogbn-arxiv and 7.8% on ogbn-products. Across all three large-scale datasets, MoLE-GNN consistently outperforms prior PEFT approaches, including AdapterGNN and GCNConv-Adapter, when built upon the NodeFormer pre-training backbone. Furthermore, as shown in Table 3, using DIFFormer-s as the backbone, MoLE-GNN surpasses both full fine-tuning and other PEFT baselines on every large-scale dataset. These results demonstrate that MoLE-GNN is not only effective on small-scale transductive benchmarks but also scales reliably to large-scale datasets, thereby confirming its scalability.

**Link Prediction Results.** We evaluate the performance of MoLE-GNN on link prediction task, with results summarized in Table 6. For this task, each expert in MoLE-GNN is instantiated as NAGphormer Chen et al. (2022a). Across all datasets, MoLE-GNN consistently outperforms fully fine-tuned NAGphormer, achieving MRR scores of 49.3%, 66.4%, and 44.7% on Cora, Citeseer, and PubMed, compared to 8.9%, 13.1%, and 7.3% from fine-tuning. We further observe that MoLE-GNN achieves a Hit@20 score of 93.2%, surpassing the full fine-tuning result of 83.3%. Again, we observe that MoLE-GNN also outperforms the MoE-based model Link-MoE on Cora, Citeseer, and ogbl-ddi, while achieving comparable performance on PubMed. While Link-MoE trains experts from scratch, our model tunes only selected components yet achieves better results, which is particularly noteworthy. MoLE-GNN consistently outperforms prior methods: it achieves MRR scores of 49.3% and 66.4% (vs. 44.0% and 64.6% for Link-MoE), and Hits@20 of 93.2% on

ogbl-ddi (vs. 85.2%). On PubMed, MoLE-GNN attains 44.7% MRR, close to Link-MoE's 53.1%. Moreover, MoLE-GNN yields average improvements over graph-specific PEFT baselines (50.9% vs. GCNconv-Adapter and 71.2% vs. AdapterGNN), while also surpassing classical PEFT techniques.

**MoLE-GNN Vs General Purpose MoE models.** In this section, we compare MoLE-GNN with SOTA general-purpose graph MoE models, as illustrated in Fig. 3. We consider GMoE and DA-MoE as baselines. From Fig. 3(a), MoLE-GNN consistently outperforms GMoE and DA-MoE on HIV, Tox21, and BBBP in terms of ROC-AUC, and achieves performance comparable to DA-MoE on ToxCast for inductive molecular graph classification. Specifically, MoLE-GNN surpasses the second-best model, DA-MoE, by 0.9%, 1.1%, and 7.8% on HIV, Tox21, and BBBP, respectively, yielding an average improvement of 2.4% across the four datasets. From Fig. 3(b), MoLE-GNN also consistently outperforms GMoE and DA-MoE on IMDB-B, IMDB-M, and COLLAB, while achieving performance comparable to DA-MoE on RDT-M for inductive social network graph classification. In particular, MoLE-GNN exceeds DA-MoE by 0.6%, 7.0%, and 3.4% on IMDB-B, IMDB-M, and COLLAB, respectively, with an average gain of 1.4% across the four datasets.

Finally, Fig. 3(c) shows that MoLE-GNN achieves superior performance over both baselines on four transductive node classification benchmarks: Cora, Citeseer, PubMed, and Texas. Relative to DA-MoE, MoLE-GNN improves Accuracy by 3.3%, 6.6%, 1.9%, and 25.5% on Cora, Citeseer, PubMed, and Texas, respectively, resulting in an average margin of 7.96% across all four datasets. Finally, for link prediction tasks, we observe from Fig. 3(d) shows that MoLE-GNN outperforms GMoE and DA-MoE on four datasets: Cora, Citeseer, PubMed, and ogbl-ddi. Compared to DA-MoE, MoLE-GNN improves the MRR by 51.6%, 36.7%, and 157.2% on Cora, Citeseer, and PubMed, respectively, and boosts Hits@20 by 104.5% on ogbl-ddi. For vanilla scratch GNNs, MoLE-GNN consistently outperforms across all datasets and for all three tasks—inductive, transductive, and link prediction as shown in Fig. 3.

**Efficiency Analysis.** We compare the per-epoch training time and the number of trainable parameters between MoLE-GNN and several baseline tuning strategies, including full fine-tuning, adapter-based tuning, and MoE-based tuning. All per-epoch times are measured on a NVIDIA A6000 GPU server, and for fairness, all methods are evaluated using the AttrMasking pretraining setup. From Table 4, we observe that MoLE-GNN achieves the lowest per-epoch training time while requiring the fewest trainable parameters across all fine-tuning paradigms. Moreover, Table 1 shows that MoLE-GNN not only provides a significantly smaller parameter footprint compared to full fine-tuning, but also delivers superior predictive performance and also same for the adapter-tuning and MoE based tuning categories. In addition, we have also reported wall clock time for each graph based MoE methods in the Appendix (Table 14).

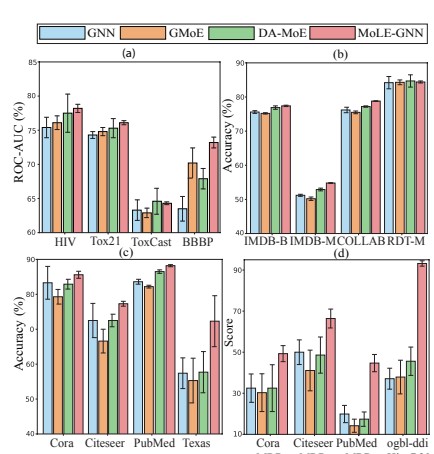

Figure 3: Performance comparison on graph classification for (a) molecular graphs and (b) social network graphs in inductive settings, (c) node classification in the transductive setting, and (d) link prediction, against state-of-the-art MoE models, the base GNN based model, and our model MoLE-GNN. Standard deviations are represented as error bars on top of the corresponding bar plots.

| Form | Bace | BBBP | ClinTox | SIDER | BN | Avg |
|---|---|---|---|---|---|---|
| Sequential (After MLP) | 70.7± 1.8 | 65.8± 0.6 | 68.4 ± 2.3 | 61.3 ± 0.5 | ✓ | 66.6 |
| Parallel (Before MP) | 80.9 ± 0.6 | 72.4 ± 0.5 | 77.1 ± 1.1 | 62.3 ± 0.4 | ✓ | 73.2 |
| Parallel (After MP) | 81.3 ± 0.7 | 72.6 ± 0.9 | **80.9** ± 0.5 | 62.6 ± 0.7 | ✓ | 74.3 |
| Parallel (Dual) | 80.6 ± 1.2 | 72.0 ± 0.7 | 71.2 ± 3.9 | 61.9 ± 0.5 | × | 71.4 |
| Parallel (Dual) | **81.6** ± 0.8 | **73.2** ± 0.8 | 80.0 ± 1.5 | **62.8** ± 0.6 | ✓ | **74.4** |

Table 7: Comparison of different adapter forms with and without BatchNorm (BN). The best result is in **bold**, and the second best model is underlined.

| Scaling | Bace | BBBP | ClinTox | SIDER | Avg |
|---|---|---|---|---|---|
| 0.01 | 79.6 ± 1.4 | 70.9 ± 0.6 | 74.7 ± 1.4 | 61.8 ± 0.7 | 71.8 |
| 0.1 | 81.2 ± 0.9 | 72.5 ± 0.6 | 79.3 ± 1.8 | 62.4 ± 0.7 | 73.9 |
| 0.5 | 81.4 ± 0.8 | 72.9 ± 0.5 | 79.5 ± 1.8 | 62.7 ± 0.4 | 74.1 |
| 1 | **81.9** ± 0.6 | 72.4 ± 0.6 | 78.9 ± 1.8 | 62.6 ± 0.3 | 73.9 |
| 5 | 81.2 ± 1.7 | 71.6 ± 0.9 | 74.8 ± 2.5 | 61.8 ± 0.5 | 72.3 |
| 10 | 80.8 ± 1.3 | 71.1 ± 1.2 | 73.9 ± 4.2 | 61.5 ± 0.9 | 71.8 |
| Learnable | 81.6 ± 0.8 | **73.2** ± 0.8 | **80.0** ± 1.5 | **62.8** ± 0.6 | **74.4** |

Table 8: Performance comparison between learnable scaling and fixed scaling. The best performing result is in **bold**, and the second best model is underlined

## 6 ABLATION STUDY

In this section, we analyze the core design choices of our proposed adapter within each expert of MoLE-GNN. We first evaluate the impact of different adapter variants, considering the presence or absence of batch normalization (BN) and the use of learnable versus fixed scaling. For MoE component of MoLE-GNN, we investigate the effect of structure-based gating and further examine the role of PEFT within the MoE framework. For brevity, additional analyses are provided in Appendix G.

**Impact of Insertion Form and BN.** Here, we analyze the impact of different insertion strategies and the effect of BN in the adapters integrated into each expert of the GNNs within the MoLE-GNN framework. Each expert contains dual adapters placed in parallel with the GNN MLP, positioned before and after message passing. To examine the effectiveness of this architecture, we compare it against variants using a single parallel adapter and a sequential adapter inserted after the GNN MLP. From Table 7, we observe that using two parallel adapters yields better performance for MoLE-GNN compared to a single adapter per expert. Moreover, omitting BN in each adapter within an expert leads to a significant performance drop.

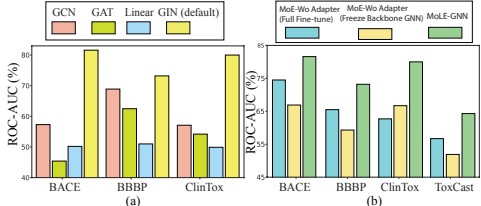

**Scaling Strategy.** We compare our proposed learnable scaling strategy against fixed scaling values ranging from 0.01 to 5, applied to each adapter within every expert of MoLE-GNN. Table 8 demonstrates that our learnable scaling yields the best results on five out of six datasets and also achieves the highest overall average performance. We observed that as the scaling factor increased, performance degraded due to catastrophic forgetting of pretrained knowledge.

Figure 4: (a) Ablation study on the gating network, and (b) the effect of PEFT within the MoE framework. Specifically, we compare the performance of a fully fine-tuned MoE framework, a frozen MoE framework, and a PEFT-based MoLE-GNN MoE framework.

**Impact of Structure-Based Gating Network.** As discussed earlier in Section 3, our gating network employs a GNN model that incorporates structural information, in contrast to a gating network based on linear projections. Fig. 4 (a) presents the results of the study conducted to further verify the effectiveness of the structure-based gating network. From Fig. 4 (a), we observe that incorporation of the gating network with the GNN backbone experts consistently yields better performance than using a linear projection. Specifically, a noticeable drop in performance is observed when the structure-based gating network is removed (i.e., 62.4% on BACE, 43.5% on BBBP, and 60.4% on ClinTox).

**Impact of PEFT in MoE framework.** We perform ablation studies to evaluate the impact of applying PEFT to pre-trained expert GNNs within our MoE-based architecture. Specifically, we compare PEFT against two baselines: full fine-tuning of the expert GNNs, and freezing the experts while tuning only the linear prediction layer. As illustrated in Fig. 4 (b), MoLE-GNN, which leverages PEFT for each expert GNN, consistently achieves better performance across all tasks, demonstrating the effectiveness of parameter-efficient fine-tuning in this setting. Specifically, a noticeable drop in performance is observed when PEFT is not applied to the pre-trained experts such as 15.8% on BACE, 17.6% on BBBP, 23.8% on ClinTox, and 18.6% on ToxCast when compared to both full fine-tuning of the expert GNNs and the strategy of freezing the experts while tuning only the linear prediction layer. Full fine-tuning may perform poorly due to catastrophic forgetting and overfitting, as updating all expert GNN parameters can erase learned representations Goodfellow et al. (2013).

## 7 CONCLUSION

In this study, we propose MoLE-GNN, an effective MoE-based PEFT framework specifically designed for GNNs. Our approach addresses the *depth-sensitivity* issue inherent in traditional fine-tuning strategies, while significantly reducing the number of tunable parameters. MoLE-GNN utilized different pre-trained adapter GNN layers as experts and allowed each individual graph to adaptively select experts. This framework highlights two key features: the structure based gating network and pre-trained GNN experts. Through comprehensive experiments on graph, node classification, and link prediction tasks, MoLE-GNN demonstrates strong generalization capabilities across datasets of varying scales. Future directions for extending our work include incorporating pre-trained graph transformers as experts, which may enhance the model's representational capacity. Additionally, MoLE-GNN could be extended to the domain of graph self-supervised learning.

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

# MoLE-GNN: Mixture of Expert Based Parameter-Efficient Fine-Tuning for Graph Neural Networks (Technical Appendix)

## The Use of Large Language Models (LLMs)

No Large Language Models (LLMs) were used in conducting the research presented in this paper. However, we employed an LLM (ChatGPT) solely for editorial purposes, including refining grammar, spelling, word choice, and overall clarity of the manuscript.

## A  Related Work

**Graph Neural Networks.** GNNs are designed to process graph-structured data through message passing to update node representations. State-of-the-art (SOTA) models include GatedGCN Li et al. (2016), GCN Kipf & Welling (2017), GAT Veličković et al. (2018), GIN Xu et al. (2019), and GraphSAGE Hamilton et al. (2017a). Recent advances focus on enhancing information aggregation, such as SAGIN Zeng et al. (2023) multi-level subgraph encoding and CF-GNNs Huang et al. (2024) conformal prediction. However, most GNNs use a fixed layer depth, limiting their adaptability to graphs of varying scales and complexities.

**Pre-trained GNN Models.** Inspired by the success of pre-trained models in NLP and vision, recent research has increasingly focused on pre-trained GNNs Qiu et al. (2020); Long et al. (2022); Xia et al. (2022b). These methods use self-supervised learning to extract meaningful representations from large-scale pre-training graphs. GAE Kipf & Welling (2016) learns via edge prediction, while DGI Velickovic et al. (2019) and InfoGraph Sun et al. (2019) maximize mutual information between graph-level and substructure-level representations. Hu *et al.* Hu et al. (2020a) use attribute masking and context prediction for molecular and protein property pre-training. GROVER Rong et al. (2020) and MGSSL Zhang et al. (2021) focus on motif prediction to leverage molecular domain knowledge. Graph contrastive learning methods like GraphCL You et al. (2020) and JOAO You et al. (2021) employ diverse augmentations for effective representation learning. However, existing methods face challenges like catastrophic forgetting, overfitting with limited data, and fixed layer configurations that fail to adapt to graph scale variations.

**Mixture of Expert Models.** The Mixture-of-Experts (MoE) framework, originally introduced by Jacobs *et al.* Jacobs et al. (1991); Jordan & Jacobs (1994), involves training specialized expert networks. Aljundi *et al.* Aljundi et al. (2017) and Shazeer *et al.* Shazeer et al. (2017) propose expert selection via auto-encoder and sparse gating, respectively. MoE modules are now widely used in computer vision Dai et al. (2021); Yu et al. (2024) and NLP Fedus et al. (2022); Du et al. (2022). In recent times, the integration of MoE techniques with GNNs has gained prominence. For example, TopExpert Kim et al. (2023) employs clustering-based gating, GMoE Wang et al. (2023) captures multi-hop information, and G-FAME Liu et al. (2023c) emphasizes fairness. Link-MoE Ma et al. (2024) and GraphMETRO Wu et al. (2023b) tackle task specialization and distribution shifts, while DA-MoE Yao et al. (2024) adapts the depth of the GNN to adapt graph scale variations. However, most MoE-GNNs are trained from scratch with a large number of parameters and do not use pre-trained GNNs as experts.

**Graph-Prompt Tuning Methods.** Prompt tuning methods, originating in NLP, adapt pre-trained models to downstream tasks by modifying inputs rather than model architecture Liu et al. (2021a); Lester et al. (2021). Variants include prefix-tuning Li & Liang (2021), which updates task-specific parameters per layer; adapter tuning Houlsby et al. (2019); Chen et al. (2022b), which inserts bottleneck adapters; BitFit Zaken et al. (2021), which tunes only bias terms; and LoRA Hu et al. (2022), which uses low-rank decomposition. These techniques have also been increasingly adopted in GNNs Wu et al. (2023c). GPPT Sun et al. (2022) introduces a framework tailored to node-level tasks, while MoLCPT Diao et al. (2022) targets molecular graphs by embedding motif information. GPF Fang et al. (2022) and GraphPrompt Liu et al. (2023b) improve parameter efficiency but struggle to match full fine-tuning performance in standard settings. Recently, AdapterGNN Li et al. (2024) extends adapter-based tuning to GNNs by integrating lightweight adapters into each layer, enabling efficient adaptation with minimal parameter updates. Despite these advances, existing parameter-efficient methods use a fixed layer configuration for all graphs, limiting their ability to adapt to varying data scales within a dataset. To overcome this, we propose an adapter-based MoE framework that employs pre-trained GNNs as experts within a MoE architecture. By integrating

tunable adapters into each expert, our approach addresses key challenges such as *depth sensitivity*, catastrophic forgetting, and overfitting, offering a more flexible and scalable fine-tuning strategy.

## B  DATASETS

Here, we provide a detailed description of the pre-training datasets and downstream tasks used in our experiments.

**Pre-training Datasets.** For pre-training our GNN experts, we use three domains of unlabeled data: two million molecules from ZINC15 Sterling & Irwin (2015), 395K protein ego-networks from PPI data, and general graph datasets grouped into academic and social categories for our inductive learning experiments. The academic set includes NetRep Ritchie et al. (2016) and two DBLP datasets from SNAP Leskovec & Sosič (2016) and NetRep, while the social set comprises Facebook and IMDB graphs from NetRep and the LiveJournal dataset from SNAP. Table 9 provides detailed statistics for each datasets. In pre-training for transductive learning (i.e., node classification), we use the ogbn-arxiv dataset from the Open Graph Benchmark (OGB) Hu et al. (2020a), a large-scale citation network comprising over 169K computer science papers (nodes) and 1.1M citation links (edges). For link prediction tasks, we likewise use ogbn-arxiv to pre-train the GNN experts.

| Dataset | Academia | DBLP(SNALP) | DBLP(NetRep) | IMDB | Facebook | LiveJournal |
|---------|----------|-------------|--------------|------|----------|-------------|
| $|V|$ | 137,969 | 317,080 | 540,486 | 896,305 | 3,097,165 | 4,843,953 |
| $|E|$ | 739,984 | 2,099,732 | 30,491,158 | 7,564,894 | 47,334,788 | 85,691,368 |

Table 9: Statistics of datasets for pre-training on general graphs.

**Downstream Tasks Datasets.** For inductive graph classification settings, we use eight molecular property prediction datasets Hu et al. (2020b): BACE (1.5K), BBBP (2.0K), ClinTox (1.4K), HIV (41K), SIDER (1.4K), Tox21 (7.8K), MUV (93K), and ToxCast (8.5K). We categorize BACE, BBBP, ClinTox, and SIDER as small-scale datasets; Tox21 and ToxCast as medium-scale; and HIV and MUV as large-scale. In addition, we include the PPI dataset (88K) Hu et al. (2020b), which is also large-scale. Following prior work Hu et al. (2020b), we adopt the *scaffold split* Ramsundar et al. (2019) for all molecular graph datasets and the *species split* for biological datasets. For transductive node classification, we evaluate MoLE-GNN on five standard benchmarks: Cora, Citeseer, PubMed Sen et al. (2008), Wisconsin, and Texas Pei et al. (2019); Tang et al. (2009). The first three are citation networks, where nodes represent documents and edges denote citation links; these datasets are homophilic. The latter two are webpage hyperlink networks, where nodes are webpages and edges correspond to hyperlinks; these datasets are heterophilic. We use a standard random 50%,25%, and 25% split for train/val/test. For link prediction, we evaluate on four standard benchmarks: Cora, Citeseer, PubMed, and ogbl-ddi. Cora, Citeseer, and PubMed are smaller graphs, whereas ogbl-ddi is substantially larger with more nodes and edges. We follow fixed train/validation/test splits of 85%, 5%, and 10% for the first three datasets, and use the official splits provided by the OGB benchmark Hu et al. (2020a) for ogbl-ddi. For the large-scale datasets ogbn-arxiv, ogbn-proteins, and ogbn-products, we follow the official train/validation/test splits provided by the OGB benchmark Hu et al. (2020a). A detailed description of all downstream task datasets is provided in Table 10.

## C  BASELINE MODELS

We compare the results of our model MoLE-GNN against fully fine-tuned GNNs. We first describe the pre-training and fine-tuning based models, along with prompt-based and adapter-based approaches. Subsequently, we provide a brief overview of MoE-based models employed in the context of GNNs.

**Pre-train & Fine-tune based Learning Methods.** Recently, researchers have explored transfer learning for GNNs, where models are pre-trained with self-supervised or unsupervised objectives and then fine-tuned on downstream tasks. Hu *et al.* Hu et al. (2020b) introduced pre-training strategies such as EdgePred and AttrMasking, followed by full fine-tuning on molecular and biological property prediction datasets in inductive settings. Similarly, for inductive learning experiments on

| Dataset | Graphs | Avg.nodes | Avg.edges | Features | Node classes | Task (I / T / L) | Category |
|---|---|---|---|---|---|---|---|
| Cora | 1 | 2,708 | 5,429 | 1,433 | 7 | T/L | Homophilic |
| PubMed | 1 | 19,717 | 88,648 | 500 | 3 | T/L | Homophilic |
| CiteSeer | 1 | 3,327 | 9,104 | 3,703 | 6 | T/L | Homophilic |
| Wisconsin | 1 | 251 | 515 | 1703 | 5 | T | Heterophilic |
| Texas | 1 | 183 | 325 | 1703 | 5 | T | Heterophilic |
| ogbn-arxiv | 1 | 169,343 | 1,166,243 | 128 | 40 | T | Large-scale |
| ogbn-proteins | 1 | 132,534 | 39,561,252 | 8 | 2 | T | Large-scale |
| ogbn-products | 1 | 2,449,029 | 61,859,140 | 100 | 47 | T | Large-scale |
| ogbl-ddi | 1 | 4,267 | 1,334,889 | 0 | – | L | – |

| Dataset | Graphs | Avg.nodes | Avg.edges | Features | Graph classes | Task (I / T / L) | Domain |
|---|---|---|---|---|---|---|---|
| BACE | 1513 | 34.1 | 73.7 | 18 | 1* | I | small molecule |
| BBBP | 2039 | 24.1 | 51.9 | 23 | 1* | I | small molecule |
| ClinTox | 1477 | 26.2 | 55.8 | 38 | 2* | I | small molecule |
| HIV | 41127 | 25.5 | 54.9 | 61 | 1* | I | small molecule |
| SIDER | 1427 | 33.6 | 70.7 | 50 | 27* | I | small molecule |
| Tox21 | 7831 | 18.6 | 38.6 | 61 | 12* | I | small molecule |
| MUV | 93087 | 24.2 | 52.6 | 15 | 17* | I | small molecule |
| ToxCast | 8576 | 18.8 | 38.5 | 63 | 617* | I | small molecule |
| PPI | 88000 | 49.4 | 890.8 | 10 | 40* | I | proteins |
| IMDB-B | 1000 | 19.8 | 96.5 | 0 | 2 | I | social network |
| IMDB-M | 1500 | 13.0 | 65.9 | 0 | 3 | I | social network |
| COLLAB | 5000 | 74.5 | 2457.2 | 0 | 3 | I | social network |
| RDT-B | 2000 | 429.6 | 497.8 | 0 | 2 | I | social network |
| RDT-M | 5000 | 508.5 | 594.9 | 0 | 5 | I | social network |

Table 10: Statistics of all datasets. Settings: T—transductive (node classification), I—inductive (graph classification), and L—link prediction. An asterisk (*) indicates the number of prediction tasks.

social network graphs, we consider the pre-training strategies proposed by Qiu *et al.* Qiu et al. (2020), namely GCC (E2E) and GCC (MoCo). For inductive learning, we employ two pre-trained graph transformers, NodeFormer Wu et al. (2022) and DIFFormer-s Wu et al. (2023a), as adapter-based baselines, and a pre-trained graph convolutional network (GCN) as the prompt-based baseline. For link prediction, we used the pre-trained NAGphormer Chen et al. (2022a) as the fine-tuning baseline.

**Graph Prompt and Adapter based Learning Methods.** Here, we consider GPF Fang et al. (2022) as the graph prompt based method to compare against our method MoLE-GNN. We did not consider other related works such as GPPT Sun et al. (2022) and GraphPrompt Liu et al. (2023b), as they are either inefficient or fail to deliver satisfactory performance without a few-shot setting for graph classification tasks, as observed by Li *et al.* Li et al. (2024). For node classification task (transductive learning), we consider the prompt tuning methods presented in ProG Zi et al. (2024), including GPPT Sun et al. (2022), All-in-one Zi et al. (2024), GraphPrompt Liu et al. (2023b), and GPF, as baseline prompt tuning-based methods. We consider the state-of-the-art PEFT models, namely $(IA)^3$ Liu et al. (2022), BitFit Zaken et al. (2021), LoRA Hu et al. (2021), Adapter Chen et al. (2022b), AdapterGNN Li et al. (2024), and GConv-Adapter Papageorgiou et al. (2025) as baseline Adapter based models to compare our MoE-based model MoLE-GNN. Additionally, we adopt domain-specific MoE models, including TopExpert Kim et al. (2023) for inductive learning and Link-MoE for link prediction, as baselines.

**MoE-based Learning Methods.** Here, we consider two state-of-the-art MOE-based GNN models, GMoEWang et al. (2023) and DA-MoEYao et al. (2024), as baseline models for comparing with our model, MoLE-GNN. GMoE employs multiple experts within each layer, while DA-MoE uses a dynamic MoE-based technique to capture information from the input graph.

# D   DETAILS OF PRE-TRAINED GNN MODELS

**Pre-Trained Models: Inductive** For inductive learning experiments, we employ a five-layer Graph Isomorphism Network (GIN) backbone Hu et al. (2020b). On molecular graphs, we evaluate seven pre-training strategies. EdgePred Hamilton et al. (2017b) masks and reconstructs edges to predict their existence, while AttrMasking and ContextPred Hu et al. (2020b) focus on attribute prediction and structural context, respectively. GraphCL You et al. (2020) introduces a contrastive framework for unsupervised graph representation learning, and SimGRACE Xia et al. (2022a) leverages a GNN and its perturbed counterpart as dual encoders to generate correlated views without requiring data augmentation. GCC (E2E) and GCC (MoCo) Qiu et al. (2020) employ self-supervised techniques to learn universal network properties across multiple graphs. For all the pre-trained models, we utilized the default checkpoints provided in their official repositories.

**Pre-Trained Models: Transductive** For the transductive learning experiments, we utilized the NodeFormer Wu et al. (2022) and DIFFormer-s Wu et al. (2023a) GT architectures. Both models are pre-trained using the ogbn-arxiv dataset from the Open Graph Benchmark (OGB) Hu et al. (2020b). Pre-training on this large, structurally diverse dataset enables the models to learn expressive node representations and capture key structural patterns in citation networks, thereby enhancing their transferability when fine-tuned on downstream tasks. For NodeFormer, we set the model with 32 hidden channels, 5 layers, and a single attention head, employing an identity transformation for the relation bias and a regularization weight of 0.1. The training employed Gumbel-Softmax sampling to enhance message passing, alongside batch normalization and residual connections. Optimization was performed with a learning rate of 0.01, no weight decay, and a batch size of 10,000 over 100 epochs. Similarly, DIFFormer-s was pre-trained with 64 hidden dimensions across 5 layers, employing a single attention head and setting the residual balance parameter $\alpha$ to 0.5. We adopt batch normalization and residual connections, together with graph positional embeddings. The attention mechanism uses a simple kernel in which queries and keys are normalized before computing dot-product attention. Training is regularized with a dropout rate of 0.2 and weight decay of 0.01, using a learning rate of 0.001, a batch size of 10,000, and 1,000 training epochs. For both NodeFormer and DIFFormer-s, we follow the original papers Wu et al. (2022; 2023a) for all hyperparameters and architectural settings, ensuring consistency with their implementations. For graph prompt-based benchmarks, we adopt DGI Veličković et al. (2018) and GraphMAE Hou et al. (2022). DGI maximizes mutual information between node- and graph-level representations to learn informative embeddings, while GraphMAE reconstructs masked features to capture deep node representations. We use Prog Zi et al. (2024) as an open-source framework to obtain the pre-trained models for both methods.

**Pre-Trained Models: Link Prediction** For link prediction experiments, we adopt NAGphormer Chen et al. (2022a) as the underlying graph transformer architecture. The model is pre-trained with the GraphMAE strategy Hou et al. (2022), from which we extract node embeddings. We employ the ogbn-arxiv dataset from OGB, a large-scale social network benchmark, aligning with our downstream tasks that also focus on social such as Cora, Citeseer, and PubMed. We configure the NAGphormer model with a hidden dimension of 128, 3 layers, 3 hops, and 8 attention heads. For NAGphormer, we adopt the hyperparameter and architectural configurations from the original paper Chen et al. (2022a) to ensure consistency with its implementation.

# E   IMPLEMENTATION DETAILS OF MOLE-GNN

All our experiments are performed on computing servers equipped with NVIDIA A6000 (48GB) and NVIDIA A100 (80GB) GPUs. We train our MoLE-GNN with freezing backbone GNN models for 100 epochs both for graph and node classification tasks. We consider Adam Kingma & Ba (2014), batch size of 256 and learning rate of 0.001. We run our MoLE-GNN model ten times with different random seeds and report the mean and standard deviation of the obtained ROC-AUC scores and accuracy to demonstrate the consistency of the results. We use PyTorch and PyG Fey & Lenssen (2019) to conduct all experiments in this work. For the adapter hyperparameter used in each expert GNN model, the bottleneck dimension is set to 15, and the initial value of the learnable scaling parameter is 0.01.

| Pre-training Method | Tuning Method | Datasets (Accuracy ↑) | | | | | |
|---|---|---|---|---|---|---|---|
| | | IMDB-B | IMDB-M | COLLAB | RDT-B | RDT-M | Avg. |
| GCC (E2E) | Full Fine-tune (100%) | $72.9_{\pm 0.03}$ | $47.9_{\pm 0.02}$ | $76.5_{\pm 0.02}$ | $83.2_{\pm 0.01}$ | $49.8_{\pm 0.02}$ | $66.1$ |
| | GPF (0.80%) | $69.4_{\pm 0.03}$ | $45.7_{\pm 0.02}$ | $\mathbf{79.8}_{\pm 0.01}$ | $73.3_{\pm 0.03}$ | $47.6_{\pm 0.03}$ | $63.2$ |
| | AdapterGNN (26.7%) | $72.6_{\pm 0.03}$ | $47.4_{\pm 0.02}$ | $72.2_{\pm 0.02}$ | $82.4_{\pm 0.03}$ | $47.1_{\pm 0.03}$ | $64.3$ |
| | **MoLE-GNN (ours) (8.3%)** | $\mathbf{76.9}_{\pm 0.01}$ | $\mathbf{54.7}_{\pm 0.01}$ | $78.5_{\pm 0.01}$ | $\mathbf{85.5}_{\pm 0.02}$ | $\mathbf{51.6}_{\pm 0.01}$ | $\mathbf{69.4}$ |
| GCC (MoCo) | Full Fine-tune (100%) | $74.5_{\pm 0.02}$ | $52.2_{\pm 0.01}$ | $79.8_{\pm 0.01}$ | $82.9_{\pm 0.01}$ | $49.9_{\pm 0.02}$ | $67.8$ |
| | GPF (0.80%) | $73.9_{\pm 0.01}$ | $50.9_{\pm 0.02}$ | $\mathbf{80.0}_{\pm 0.01}$ | $83.0_{\pm 0.01}$ | $51.1_{\pm 0.02}$ | $67.8$ |
| | AdapterGNN (26.7%) | $74.4_{\pm 0.03}$ | $50.8_{\pm 0.03}$ | $79.0_{\pm 0.01}$ | $83.7_{\pm 0.02}$ | $45.6_{\pm 0.02}$ | $66.7$ |
| | **MoLE-GNN (ours) (8.3%)** | $\mathbf{77.4}_{\pm 0.02}$ | $\mathbf{54.8}_{\pm 0.01}$ | $78.8_{\pm 0.01}$ | $\mathbf{84.4}_{\pm 0.03}$ | $51.4_{\pm 0.01}$ | $\mathbf{69.3}$ |

Table 11: Test Accuracy (%) performances on graph classification benchmarks with different tuning methods and pre-trained GNN models. Results are reported as mean ± standard deviation of Accuracy. The best result is in **bold**, and the second best model is underlined.

| Pre-training Method | Tuning Method | Datasets (ROC-AUC ↑) | | | | | |
|---|---|---|---|---|---|---|---|
| | | Cora | Citesser | Pubmed | Wisconsin | Texas | Avg. |
| DGI | Full Fine-tune (100%) | $78.7_{\pm 7.1}$ | $85.0_{\pm 3.1}$ | $\mathbf{93.2}_{\pm 0.4}$ | $58.7_{\pm 4.9}$ | $48.2_{\pm 4.7}$ | $72.7$ |
| | GPPT (3.8%) | $50.0_{\pm 1.0}$ | $50.0_{\pm 1.6}$ | $50.2_{\pm 0.1}$ | $65.2_{\pm 3.2}$ | $56.2_{\pm 0.1}$ | $54.3$ |
| | All-in-one (5.7%) | $47.9_{\pm 1.6}$ | $53.4_{\pm 5.5}$ | $49.4_{\pm 2.2}$ | $73.5_{\pm 7.4}$ | $67.7_{\pm 7.4}$ | $58.4$ |
| | GraphPrompt (0.01%) | $70.1_{\pm 0.7}$ | $55.9_{\pm 0.8}$ | $50.8_{\pm 0.2}$ | $71.9_{\pm 4.8}$ | $56.4_{\pm 3.1}$ | $61.0$ |
| | GPF (0.57%) | $48.4_{\pm 0.01}$ | $59.9_{\pm 0.1}$ | $76.3_{\pm 0.1}$ | $89.0_{\pm 1.0}$ | $69.8_{\pm 0.2}$ | $68.7$ |
| | **MoLE-GNN (ours) (6.4%)** | $\mathbf{97.2}_{\pm 0.2}$ | $\mathbf{97.7}_{\pm 0.6}$ | $92.8_{\pm 0.3}$ | $\mathbf{95.8}_{\pm 8.0}$ | $\mathbf{87.9}_{\pm 9.8}$ | $\mathbf{94.3}$ |
| GraphMAE | Full Fine-tune (100%) | $\mathbf{86.8}_{\pm 2.6}$ | $\mathbf{87.0}_{\pm 0.6}$ | $\mathbf{93.5}_{\pm 0.3}$ | $57.9_{\pm 6.8}$ | $50.2_{\pm 4.9}$ | $75.1$ |
| | GPPT (3.8%) | $56.5_{\pm 1.6}$ | $60.0_{\pm 0.8}$ | $85.7_{\pm 0.7}$ | $63.3_{\pm 8.6}$ | $49.6_{\pm 0.03}$ | $63.0$ |
| | All-in-one (5.7%) | $56.3_{\pm 0.9}$ | $67.6_{\pm 1.2}$ | $85.2_{\pm 0.7}$ | $69.2_{\pm 6.7}$ | $51.8_{\pm 10.6}$ | $66.0$ |
| | GraphPrompt (0.01%) | $60.2_{\pm 2.6}$ | $60.9_{\pm 1.2}$ | $82.9_{\pm 1.0}$ | $66.8_{\pm 8.8}$ | $48.9_{\pm 9.2}$ | $63.9$ |
| | GPF (0.57%) | $81.5_{\pm 2.1}$ | $78.6_{\pm 0.6}$ | $90.1_{\pm 0.3}$ | $89.7_{\pm 7.6}$ | $63.5_{\pm 4.7}$ | $80.7$ |
| | **MoLE-GNN (ours) (6.4%)** | $68.2_{\pm 3.4}$ | $72.8_{\pm 3.7}$ | $92.2_{\pm 0.3}$ | $\mathbf{96.0}_{\pm 8.0}$ | $\mathbf{88.0}_{\pm 9.8}$ | $\mathbf{83.4}$ |

Table 12: Test performances on node classification benchmarks with different tuning methods and pre-trained GNN models. The best result is in **bold**, and the second best model is underlined.

| Strategy Name | | Fine-tuning Scenarios | | Key Properties | | | Automated |
|---|---|---|---|---|---|---|---|
| | | GNN | Graph Task | Fine-tuning Cost | Parameter Efficiency | Scalability | |
| Conventional GNNs | GCN | √ | Node/Edge/Graph | High | Low | Low | × |
| | GIN | √ | Node/Edge/Graph | High | Low | Low | × |
| | GAT | √ | Node/Edge/Graph | High | Low | Low | × |
| NAS / AutoGNNs | GraphNAS (Gao et al., 2019) | √ | Node/Graph | Very High | Moderate | Moderate | √ |
| | Auto-gnn (Zhou et al., 2022) | √ | Node/Graph | Very High | Moderate | Moderate | √ |
| Graph Foundation Models | S2PGNN (Zhili et al., 2024) | √ | Graph | Very High | Low | Low | Partial |
| | AdapterGNN (Li et al., 2024) | √ | Graph | Low | High | Low | × |
| MoE-style GNNs | DA-MoE (Yao et al., 2024) | √ | Node/Edge/Graph | Medium | Moderate | Moderately High | Partial |
| | TopExpert (Kim et al., 2023) | √ | Graph | Medium | Moderate | Moderately High | Partial |
| Proposed | MoLE-GNN | √ | Node/Edge/Graph | **Low** | **Very High** | **High** | × |

Table 13: Comparison of strategy families in terms of fine-tuning scenarios and key properties. Fine-tuning cost reflects the overall computational overhead to adapt a pre-trained GNN to downstream tasks. MoLE-GNN achieves low fine-tuning cost while maintaining very high parameter efficiency and scalability.

# F    ADDITIONAL RESULTS

We present additional results covering inductive learning on social network graphs, a comparison of MoLE-GNN with graph prompt–based methods, and the application of MoLE-GNN to few-shot learning in both inductive and transductive settings.

**Inductive Learning On Social Network Graphs.** We evaluate the performance of MoLE-GNN under inductive learning settings, with results summarized in Table 11. Across all social network datasets, MoLE-GNN consistently surpasses full fine-tuning in graph classification. Specifically, it attains 77.2% Accuracy on IMDB-B (a 4.75% gain over 73.7%), 50.1% on IMDB-M (9.38%), COLLAB (0.64%), RDT-B (2.17%), and RDT-M (3.20%), respectively. Furthermore, MoLE-GNN outperforms the graph-prompt baseline GPF on four out of five datasets, achieving an average improvement of 5.88%. Moreover, MoLE-GNN outperforms the current SOTA graph-specific PEFT method, AdapterGNN, by 5.88%, marking a substantial gain over conventional PEFT approaches.

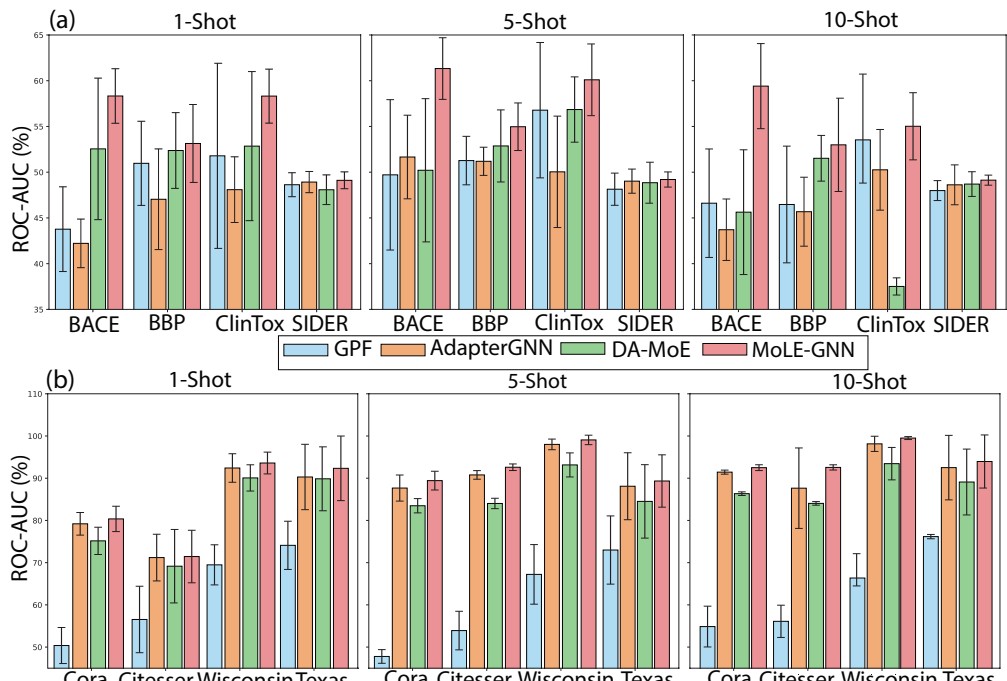

Figure 5: Performance comparison among the prompt-based method (GPF), the adapter-based method (AdapterGNN), the MoE-based method (DA-MoE), and our proposed method (MoLE-GNN) under few-shot settings for both graph classification and node classification tasks. Results are reported for 1-shot, 5-shot, and 10-shot scenarios across both tasks. Standard deviations are represented as error bars on top of the corresponding bar plots.

**Transductive Learning Results.** We evaluate the performance of MoLE-GNN under transductive learning settings against state-of-the-art graph prompt learning methods, with results summarized in Table 12. MoLE-GNN consistently achieves superior ROC-AUC performance compared to all state-of-the-art graph prompt learning methods.

**Few-shot Performance on Inductive and Transductive Learning Settings.** In recent times, prompt tuning has been well accepted for its effectiveness in addressing few-shot downstream tasks (Brown et al., 2020; Schick & Schütze, 2020b;a; Liu et al., 2021b; 2023a). GPF Fang et al. (2022) is designed for graph classification in few-shot settings but fails to generalize to node classification tasks under the same setting. Similarly, AdapterGNN Li et al. (2024) does not address either graph or node classification in few-shot scenarios. In contrast, DA-MoE Aghdam et al. (2024), a MoE-based model, does not exhibit this limitation. In this work, we perform graph and node classification tasks under few-shot settings using MoLE-GNN. For inductive (graph classification) task, we consider four small molecule datasets these are BACE, BBBP, ClinTox, and SIDER. From Fig. 5 (a) we observe that MoLE-GNN model outperforms all the baseline models by a significant improvement of 10.83% in terms of ROC-AUC score on all four inductive datasets. For transductive (node classification) tasks, we consider four datasets, these are Cora, Citeseer, Wisconsin, and Texas. From Fig. 5 (b) we observe that MoLE-GNN model outperforms all the baseline models by a margin of 1.82% on all four node classification datasets. The hybrid combination of our framework balances task-specific adaptation (via adapters) and structural flexibility (via the MoE). This synergy allows the model to focus on the most relevant depth and expert pathways without overfitting, making it robust in low-data regimes.

**Wall-Clock Efficiency and Runtime Analysis.** We measure the average per-epoch and total wall-clock time (over 100 epochs) for various MoE-based tuning strategies across six datasets (BBBP, Tox21, ToxCast, SIDER, ClinTox, and BACE). As shown in Table 14, our model, MoLE-GNN, achieves the lowest per-epoch and wall-clock times among all graph-based MoE methods, while also requiring the fewest trainable parameters compared to all baseline MoE approaches.

| Method | Per-Epoch Time (seconds) | Wall-Clock (100 epochs, Minutes) | Inference Time (seconds) | Params (M) |
|---|---|---|---|---|
| GMoE | 2.17 | 3.61 | 1.25 | 14.9 |
| DA-MoE | 1.68 | 2.80 | 0.94 | 29.8 |
| TopExpert | 1.21 | 2.02 | 0.75 | 2.51 |
| **MoLE-GNN** | **0.45** | **0.75** | **0.33** | **0.39** |

Table 14: Comparison of per-epoch, total wall-clock (100 epochs), along with trainable parameters across different MoE-based tuning strategies.

| | BACE | BBBP | ClinTox |
|---|---|---|---|
| Dense Experts | $73.1_{\pm 9.8}$ | $67.5_{\pm 5.9}$ | $76.8_{\pm 4.9}$ |
| Sparse Experts | $\mathbf{81.6}_{\pm 0.8}$ | $\mathbf{73.2}_{\pm 0.8}$ | $\mathbf{80.0}_{\pm 1.5}$ |

Table 15: Performance analysis of the different expert mechanisms used in MoLE-GNN. The best results are highlighted in **bold**. "Dense Experts" refers to the selection of all available experts in MoLE-GNN, while "Sparse Experts" indicates that only a subset of the experts is selected.

# G ABLATION STUDY

**Ablation Study on the Impact Across GNN Backbones**   Here, we evaluate the performance of MoLE-GNN using different GNN backbones within each expert, including GCN, Graph Attention Network (GAT), GraphSAGE, and our default choice, GIN. As shown in Table 16, the best performance is achieved when GIN is used as the backbone in each expert. Moreover, this configuration requires updating only a small portion of the backbone parameters while still delivering optimal results.

| Backbones in MoLE-GNN | BACE | BBBP | ClinTox | Sider | Total Params (M) | Trainable Params (M) | Trainable % |
|---|---|---|---|---|---|---|---|
| GCN | $\underline{76.7}_{\pm 2.6}$ | $66.3_{\pm 2.5}$ | $\underline{58.3}_{\pm 2.4}$ | $\underline{61.1}_{\pm 1.4}$ | 2.54M | 0.39M | 15.4 |
| GAT | $71.4_{\pm 3.1}$ | $65.5_{\pm 1.4}$ | $55.4_{\pm 5.5}$ | $60.6_{\pm 0.9}$ | 4.43M | 0.39M | 8.8 |
| GraphSAGE | $69.8_{\pm 2.8}$ | $\underline{66.4}_{\pm 3.1}$ | $55.9_{\pm 0.7}$ | $59.7_{\pm 1.7}$ | 2.54M | 0.39M | 15.4 |
| GIN | $\mathbf{81.6}_{\pm 0.8}$ | $\mathbf{73.2}_{\pm 0.8}$ | $\mathbf{80.0}_{\pm 1.5}$ | $\mathbf{62.8}_{\pm 0.6}$ | 7.7M | 0.39M | 5.1 |

Table 16: Performance analysis of different GNN backbones used in MoLE-GNN. We observe that MoLE-GNN built on top of GIN outperforms other backbones while tuning only 5.1% of the total parameters. Best performing model is **bold** and second best is underlined in terms of ROC-AUC.

**Expert Diversity: How Crucial is Heterogeneity Among Experts.**   To evaluate the importance of expert heterogeneity in MoLE-GNN, we compare configurations with homogeneous and heterogeneous experts. In the homogeneous settings, all experts share the same GNN architecture and are pre-trained on identical data. Specifically, we consider three configurations where each expert consists of (i) a 1-layer GNN, (ii) a 3-layer GNN, or (iii) a 5-layer GNN. In contrast, the heterogeneous version of MoLE-GNN employs experts with different depths and receptive fields, combining multiple GNN backbones that specialize in diverse aggregation patterns. As shown in Table 18, the heterogeneous expert design consistently achieves the best performance across all datasets, outperforming every homogeneous configuration by a notable margin (average ROC-AUC 77.9% vs. 72.2% for the best homogeneous setup). This demonstrates that structural diversity among experts is crucial to MoLE-GNN's success: varied depths and receptive fields allow different experts to capture complementary subgraph patterns, which the gating mechanism dynamically integrates for each input graph. When all experts share identical architectures and pretraining, the mixture degenerates into redundant feature extractors, limiting the benefits of the mixture-of-experts formulation. Hence, expert heterogeneity is a key factor that enhances both representational richness and generalization ability in MoLE-GNN.

| Tuning Method | BACE | BBBP | ClinTox | ToxCast | Avg. |
|---|---|---|---|---|---|
| MoLE-GNN (GNN + adapter MLP) | $75.4_{\pm3.3}$ | $66.2_{\pm3.6}$ | $69.3_{\pm4.5}$ | $63.1_{\pm0.3}$ | 68.5 |
| MoLE-GNN | $81.6_{\pm0.8}$ | $73.2_{\pm0.8}$ | $80.0_{\pm1.5}$ | $64.3_{\pm0.2}$ | **77.9** |

Table 17: Performance analysis on full fine tuning of MoLE-GNN (GNNs + adapter MLP) between MoLE-GNN, where only tunes adapter MLPs and the GNNs remain freeze, which is our proposed framework. Best performing model is **bold** and second best is underlined in terms of average ROC-AUC.

| Expert Configuration | | BACE | BBBP | ClinTox | ToxCast | Avg. |
|---|---|---|---|---|---|---|
| | 1-layer GNNs | $71.9_{\pm0.8}$ | $65.8_{\pm0.6}$ | $68.0_{\pm2.1}$ | $54.1_{\pm0.4}$ | 64.9 |
| Homogeneous Experts | 3-layer GNNs | $79.9_{\pm0.4}$ | $70.9_{\pm0.7}$ | $76.6_{\pm1.7}$ | $61.2_{\pm0.3}$ | 72.2 |
| | 5-layer GNNs | $80.8_{\pm0.3}$ | $72.4_{\pm0.5}$ | $72.9_{\pm1.5}$ | $56.9_{\pm0.6}$ | 70.8 |
| Heterogeneous Experts | MoLE-GNN | $\mathbf{81.6}_{\pm0.8}$ | $\mathbf{73.2}_{\pm0.8}$ | $\mathbf{80.0}_{\pm1.5}$ | $\mathbf{64.3}_{\pm0.2}$ | **77.9** |

Table 18: Effect of expert diversity in MoLE-GNN. Comparison between homogeneous experts (five identical GNN experts with 1, 3, or 5 layers each) and heterogeneous experts (MoLE-GNN with varied GNN backbones). Results are reported in ROC-AUC (%). Best results are highlighted in **bold**.

**Ablation Study on Full MoLE-GNN Fine-Tuning.** In this section, we evaluate the effect of full fine-tuning in MoLE-GNN, where we update the entire GNN backbone of each expert in addition to the adapters. We compare this with our default parameter-efficient variant of MoLE-GNN, in which only the adapters are trained while all expert backbones remain frozen. As shown in Table 17, the parameter-efficient MoLE-GNN consistently outperforms the full-tuning variant. We also observe that fully tuning all experts often leads to negative transfer, likely because each expert already contains a pretrained GNN backbone and overriding these pretrained weights disrupts their specialization. Furthermore, full fine-tuning requires 7.7M trainable parameters, whereas our design uses only 0.39M parameters—just 5.1% of the full-tuning model.

**Impact of Dense Expert Selection** Varying the number of top selected experts (i.e., $k$) allows the model to adaptively capture patterns across different GNN layers. As shown in Table 15, we observe that optimal performance in these datasets is achieved when MoLE-GNN employs sparse expert selection rather than using all experts densely. Specifically, compared to the dense expert configuration in MoLE-GNN, the sparse expert variant in MoLE-GNN (proposed model) achieved notable performance gains of 11.6%, 8.4%, and 4.2% across the three datasets, respectively. These improvements explain the ability of sparse experts to effectively capture aggregation information at various GNN layers.

| # Experts | BACE | BBBP | ClinTox | ToxCast |
|---|---|---|---|---|
| 1 | 62.8 | 64.4 | 56.0 | 59.1 |
| 2 | 71.6 | 65.9 | 71.6 | 61.4 |
| 3 | 75.1 | 70.6 | 75.7 | 62.5 |
| 4 | 80.4 | 72.4 | 78.1 | 63.2 |
| 5 | **81.6** | **73.2** | **80.0** | **64.3** |

Table 19: The sensitivity analysis on the choice of experts on four graph classification datasets.

**Sensitivity Analysis on the Selection of the Number of Experts** We conduct a sensitivity analysis on the number of experts used in the MoLE-GNN framework (Section 3 in main text). To this end, we performed an ablation study on four graph classification datasets. Experiments were conducted with varying numbers of expert GNNs, and the results are presented in Fig. 19. From the figure, we observe that the best ROC-AUC (%) is obtained with five experts (we cannot perform sensitivity analysis for six experts, as pre-trained GNNs can have a maximum of five layers of GNNs

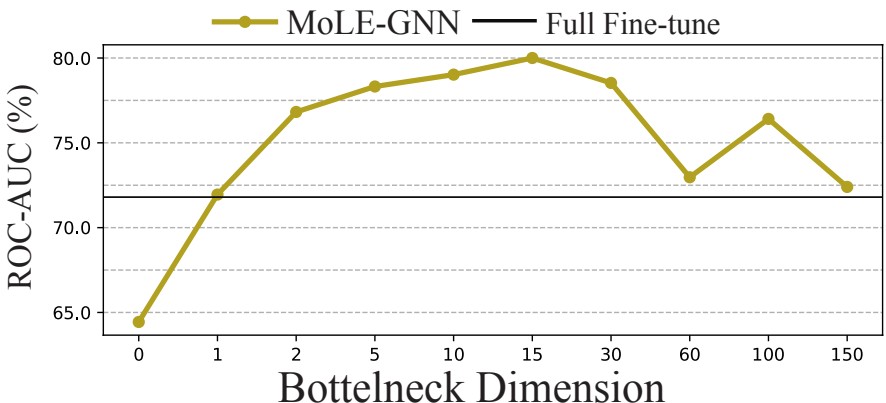

Figure 6: Performances with different bottleneck dimensions. 0 represents identical mapping. Here we consider ClinTox to perform this experiment.

| Bottleneck MLP | Datasets (ROC-AUC ↑) | | | | |
| Dim. | Bace | BBBP | ClinTox | SIDER | Avg |
| --- | --- | --- | --- | --- | --- |
| 0 | $73.1_{\pm2.4}$ | $62.8_{\pm1.2}$ | $64.4_{\pm1.3}$ | $58.4_{\pm1.7}$ | 64.7 |
| 4 | $74.6_{\pm0.9}$ | $66.0_{\pm0.5}$ | $70.1_{\pm1.8}$ | $59.4_{\pm1.5}$ | 67.5 |
| 15 | $81.6_{\pm0.8}$ | $73.2_{\pm0.8}$ | $80.0_{\pm1.5}$ | $62.8_{\pm0.6}$ | **74.4** |
| 64 | $75.2_{\pm0.9}$ | $65.2_{\pm0.5}$ | $76.3_{\pm3.5}$ | $58.4_{\pm0.4}$ | 68.8 |
| 100 | $74.2_{\pm3.6}$ | $63.2_{\pm1.4}$ | $76.4_{\pm2.0}$ | $58.3_{\pm0.7}$ | 68.1 |
| 150 | $72.8_{\pm0.4}$ | $62.6_{\pm0.2}$ | $72.4_{\pm4.0}$ | $58.7_{\pm0.5}$ | 66.6 |

(a) Graph classification tasks.

| Bottleneck MLP | Datasets (Accuracy ↑) | | | |
| Dim | Cora | Citesser | Pubmed | Avg |
| --- | --- | --- | --- | --- |
| 0 | $79.0_{\pm1.7}$ | $73.4_{\pm1.1}$ | $80.0_{\pm0.6}$ | 77.8 |
| 4 | $81.7_{\pm1.2}$ | $74.2_{\pm1.0}$ | $86.8_{\pm0.6}$ | 80.9 |
| 15 | $85.6_{\pm1.0}$ | $77.3_{\pm0.7}$ | $89.3_{\pm0.3}$ | **84.1** |
| 64 | $81.5_{\pm1.1}$ | $72.9_{\pm1.2}$ | $85.8_{\pm0.3}$ | 80.1 |
| 100 | $78.7_{\pm1.0}$ | $70.1_{\pm0.6}$ | $84.3_{\pm0.3}$ | 77.7 |
| 150 | $70.7_{\pm1.5}$ | $68.7_{\pm0.8}$ | $82.7_{\pm0.4}$ | 74.0 |

(b) Node classification tasks.

Table 20: Effect of adapter bottleneck dimension on graph classification 20a and node classification 20b tasks. Small to moderate bottleneck sizes improve performance over the no-adapter baseline, while excessively large bottlenecks reduce accuracy. Our default setting (15) consistently achieves the best average performance across all tasks, indicating that balanced adapter capacity is crucial for stable gains.

as discussed in Section D). Consequently, we selected five expert GNNs in our MoE framework MoLE-GNN for all our experiments.

**Ablation Study on Adapter MLP Capacity across different tasks.** Table 20 shows that increasing the adapter's bottleneck MLP dimension leads to overfitting on both graph classification and node classification tasks. As seen in Tables 20a and 20b, our default bottleneck dimension of 15 consistently achieves the strongest average ROC-AUC across all datasets. In contrast, setting the bottleneck size to 0 causes MoLE-GNN to underfit, while an excessively large dimension of 150 results in overfitting and degraded performance. Since the underlying GNN MLP layers belong to a pre-trained backbone and remain frozen during both training and inference, we adjust only the adapter bottleneck dimension in this ablation to control the effective capacity of the tunable parameter space.

**Bottleneck Dimension** Fig. 6 that reducing the bottleneck dimension to limit the size of tunable parameter space can improve the generalization ability of the model. But when the size of is too small, the model may suffer from underfitting, which can restrict its performance. Therefore, selecting bottleneck dimension of 15, which present 5.1% of all parameters, yields the best performance. Meanwhile, a dimension of 1, which accounts for only 0.5% of all parameters, can surpass the results of full fine-tuning.

