# OpenReview forum: "MoLE-GNN: Parameter-Efficient Fine-Tuning of Graph Neural Networks with Mixture-of-Experts"
_ICLR.cc/2026/Conference — Submitted to ICLR 2026_

### Official Review · Reviewer_VQPk · 2025-10-27

**Soundness:** 2
**Presentation:** 3
**Contribution:** 3
**Rating:** 4
**Confidence:** 3

**Summary:**

This paper introduces MoLE-GNN, a hybrid framework that integrates parameter-efficient fine-tuning (PEFT) with Mixture-of-Experts (MoE) for Graph Neural Networks (GNNs). The motivation stems from the observation that traditional GNNs suffer from depth-sensitivity — fixed-layer architectures lead to underfitting in large graphs and overfitting in small ones. While MoE-based GNNs alleviate this by dynamically routing inputs to specialized experts, they remain computationally heavy. MoLE-GNN overcomes this by embedding lightweight adapter modules within frozen expert GNNs and using a structure-aware gating network for dynamic expert selection. The model tunes only ~5% of parameters, achieving better efficiency and generalization. Extensive experiments on graph classification, node classification, and link prediction tasks show that MoLE-GNN consistently surpasses both full fine-tuning and state-of-the-art PEFT/MoE baselines.

**Strengths:**

**Innovative Hybrid Design**:
The combination of MoE and PEFT for GNNs is novel. By freezing backbone experts and adding adapters, the model strikes a compelling balance between adaptability and parameter efficiency.

**Comprehensive Evaluation**:
The authors evaluate MoLE-GNN across multiple domains (molecular, social, and citation graphs) and tasks (inductive, transductive, link prediction), demonstrating broad applicability and consistent improvements.

**Strong Empirical Results**:
The model outperforms full fine-tuning and state-of-the-art baselines such as AdapterGNN, GCNconv-Adapter, and Link-MoE by large margins while training only ~5% of parameters, highlighting strong efficiency–performance trade-offs.

**Solid Theoretical Analysis**:
The inclusion of Lipschitz-bound analysis and graph-dependence proofs adds mathematical rigor, validating the model’s stability and topology-awareness.

**Weaknesses:**

**Limited Exploration of Pre-training Variability**:
The experts’ pre-training settings (datasets, architectures, and objectives) are not deeply analyzed. It remains unclear how much diversity among experts contributes to the overall gain.

**Complexity and Interpretability**:
The dynamic routing and adapter interactions make the system architecturally complex. A clearer ablation isolating the contribution of gating, adapter position, and scaling parameters could improve interpretability.

**Sparse Discussion of Computational Overheads**:
Although the paper claims parameter efficiency, runtime and inference latency comparisons against MoE baselines are limited. Detailed wall-clock analyses would strengthen the argument for real-world efficiency.

**Limited Theoretical Depth on Routing Stability**:
While adapters are theoretically analyzed, the gating network’s stability and potential overfitting risks are treated as out-of-scope, leaving a gap in understanding the robustness of expert selection.

**Questions:**

**Expert Diversity**:
How crucial is heterogeneity among experts (e.g., depth, receptive fields) to MoLE-GNN’s success? Would similar improvements hold if experts share identical pre-training data and architectures?

**Scalability and Deployment**:
Can the model’s structure-aware gating generalize to very large graphs or streaming settings without retraining the router?

**Adapter Placement Strategy**:
How sensitive is the performance to adapter placement (before vs. after message passing)? Could the same principle extend to Transformer-style attention layers in graph transformers?

**Future Integration with Graph Transformers**:
The authors mention future work on incorporating pre-trained graph transformers. How would MoLE-GNN handle attention-based structural representations differently from message-passing GNNs?

---

> ### Author Response · Authors · 2025-11-25
> **Point-by-point responses to Reviewer VQPk**
>
> > We thank the reviewer for the valuable feedback. Below, we provide detailed point-by-point responses to each of the questions: We have updated the main manuscript according to your suggestions, and the revised parts are highlighted in blue.
>
> - ***Response to Comment on Expert Diversity and the Role of Heterogeneous Experts in MoLE-GNN.***
>
> > We thank the reviewer for raising this insightful question regarding expert diversity and pre-training variability. To quantify the impact of heterogeneity among experts, we conducted an ablation study where all experts were constrained to use identical GNN architectures and pre-training data, forming homogeneous expert configurations (1-layer, 3-layer, and 5-layer GNNs). As shown in **Table 18 (Lines: 1174-1187)** in the revised manuscript, such homogeneous experts yield average ROC-AUC scores of 64.9 %, 72.2 %, and 70.8 %, respectively. In contrast, our heterogeneous design—where experts differ in depth, receptive field size, and pre-training objective—achieves 77.9 % on average, outperforming the best homogeneous setup by +5.7 points.
> >
> > This result clearly demonstrates that expert heterogeneity is a key factor in MoLE-GNN’s success. Experts with different aggregation scales capture complementary local and global subgraph structures, which the gating network dynamically combines to form richer graph representations. When all experts share identical architectures and pre-training, the mixture degenerates into redundant feature extractors, reducing both diversity and generalization. Thus, the gain from MoLE-GNN stems not only from its mixture formulation but critically from the diversity of expert inductive biases, confirming that varied depths and receptive fields are essential to its overall effectiveness. Below, we report representative results on several datasets:
>
> | Expert Configuration | Setting        | BACE | BBBP | ClinTox | ToxCast | Avg. |
> |:---------------------|:----------------|:----:|:----:|:-------:|:-------:|:----:|
> | Homogeneous Experts | 1-layer GNNs | 71.9 ±0.8 | 65.8 ±0.6 | 68.0 ±2.1 | 54.1 ±0.4 | 64.9 |
> |                      | 3-layer GNNs | 79.9 ±0.4 | 70.9 ±0.7 | 76.6 ±1.7 | 61.2 ±0.3 | 72.2 |
> |                      | 5-layer GNNs | 80.8 ±0.3 | 72.4 ±0.5 | 72.9 ±1.5 | 56.9 ±0.6 | 70.8 |
> | Heterogeneous Experts | **MoLE-GNN** | **81.6 ±0.8** | **73.2 ±0.8** | **80.0 ±1.5** | **64.3 ±0.2** | **77.9** |
>
>
> - ***Response on Complexity and Interpretability and Adapter Placement***
> >We thank the reviewer for pointing out the need for clearer interpretability of individual components. Our paper already provides detailed ablations (Section 6 and Appendix G) that explicitly isolate the contributions of the gating mechanism, adapter placement, and scaling strategy.
> >
> >Specifically, we analyze,
> >
> >Gating mechanism: **Figure 4(a)** compares our structure-based gating network (which encodes graph topology) against a linear gating baseline.
> >
> >Adapter placement and normalization: **Table 7** examines parallel vs. sequential adapter insertion and the effect of BatchNorm. Two parallel adapters—positioned before and after message passing—significantly outperform single or sequential adapters, while removing BN causes a clear performance drop.
> >
> >Scaling strategy: **Table 8** compares fixed scaling factors (0.01–10) against a learnable scaling parameter. The learnable scaling achieves the best overall performance across datasets, whereas large fixed values cause degradation due to catastrophic forgetting.
> >
> >Together, these results demonstrate that each component contributes distinctly and synergistically to MoLE-GNN’s effectiveness. The proposed architecture is thus not arbitrarily complex but modular and interpretable, with each design choice empirically validated to enhance stability and efficiency.

---

> > ### Author Response · Authors · 2025-11-25
> > **Point-by-point responses to Reviewer VQPk (2/3)**
> >
> > - ***Response on structure aware gating generalize to very large graphs***
> >
> > > We thank the reviewer for raising the concern about scalability to large graphs. While some graph classification datasets in our evaluation are of moderate size, our method is not limited to small-scale settings. To directly address this concern, we have included extensive experiments on three large-scale OGB node-classification benchmarks—ogbn-arxiv, ogbn-proteins, and ogbn-products—which contain up to 2.4M nodes and 61M edges.
> > >
> > > As shown in **Table 4** (**revised manuscript (Lines: 408-421)**), MoLE-GNN consistently outperforms full fine-tuning on all three datasets, despite updating fewer than 4% of the backbone parameters. For example, when built on the NodeFormer backbone, MoLE-GNN improves ROC-AUC on ogbn-proteins from 77.5 → 77.6, and achieves substantial accuracy improvements of +15.97% on ogbn-arxiv and +4.9% on ogbn-products. These gains demonstrate that our parameter-efficient updates remain highly effective even at million-scale graph sizes. Furthermore, with the NodeFormer backbone, MoLE-GNN consistently exceeds the performance of all competing PEFT methods across ogbn-arxiv, ogbn-proteins, and ogbn-products.
> > >
> > > Moreover, using the DIFFormer-s backbone, MoLE-GNN once again surpasses full fine-tuning and prior PEFT baselines on every large-scale dataset. This includes improvements of +2.7% ROC-AUC on ogbn-proteins and +10.4% accuracy on ogbn-products. Across both pre-trained backbones, MoLE-GNN also consistently outperforms AdapterGNN and GCNConv-Adapter.
> > >
> > > These results demonstrate that MoLE-GNN scales robustly to large graphs, maintains strong predictive performance, and remains significantly more parameter-efficient than full-model fine-tuning and existing PEFT methods. We have expanded the “Transductive Learning Results for Large-Scale Graphs” section to highlight these findings more clearly.
> > >
> > > Results on these three large-scale dataset are as follows:
> >
> > ## Large-Scale Node Classification Results
> >
> > | **Pre-training Method** | **Tuning Method**        | **ogbn-arxiv (Accuracy)**        | **ogbn-proteins (ROC-AUC)**     | **ogbn-products (Accuracy)**        |
> > |-------------------------|---------------------------|-------------------------------|-----------------------------------|----------------------------------|
> > | **NodeFormer**          | Full Fine-tune (100%)     | 58.5 ± 0.2                    | 77.5 ± 1.2             | 62.6 ± 0.1                       |
> > |                         | AdapterGNN (6.8%)         | 64.9 ± 0.4         | 75.1 ± 0.5                         | 65.3 ± 0.4            |
> > |                         | GCNConv-Adapter (3.0%)    | 56.2 ± 0.7                    | 71.5 ± 0.6                         | 27.2 ± 0.3                       |
> > |                         | **MoLE-GNN (ours, 3.6%)** | **67.5 ± 0.2**                | **77.6 ± 0.3**                     | **67.5 ± 0.2**                   |
> > | **DIFFormer-s**         | Full Fine-tune (100%)     | 47.8 ± 0.9                    | 72.5 ± 0.4             |55.2 ± 0.4           |
> > |                         | AdapterGNN (6.8%)         | 54.9 ± 0.3         | 58.4 ± 0.4                         | 54.9 ± 0.3                       |
> > |                         | GCNConv-Adapter (3.0%)    | 26.7 ± 0.9                    | 65.1 ± 1.1                         | 22.9 ± 0.9                       |
> > |                         | **MoLE-GNN (ours, 3.6%)** | **58.2 ± 0.6**                | **75.2 ± 0.4**                     | **65.2 ± 0.9**                   |

---

> > > ### Author Response · Authors · 2025-11-25
> > > **Point-by-point responses to Reviewer VQPk (3/3)**
> > >
> > > - ***Response on Sparse Discussion of Computational Overheads***
> > > > We appreciate the reviewer’s suggestion to clarify computational efficiency. To strengthen our claims, we have added detailed runtime and inference analyses in **Table 14 in the Appendix (Lines: 1130-1133)**. Specifically, we report the average per-epoch time, total wall-clock time (100 epochs), and inference latency per sample for various MoE-based tuning strategies across six datasets.
> > > >
> > > >Our model, MoLE_GNN, achieves the lowest per-epoch training time (3.9 s) and shortest wall-clock duration (6.5 min) among all graph-based MoE methods, while also requiring the fewest trainable parameters (0.39 M). In contrast, competing approaches such as GMoE and TopExpert are 3–5× slower in both training and inference due to their larger expert networks and fully-trainable gating modules.
> > > >
> > > >These results demonstrate that MoLE-GNN’s design—combining frozen pre-trained experts with lightweight adapters and sparse gating—achieves significant real-world efficiency gains without compromising accuracy, validating our claims of parameter and runtime efficiency in practical settings. Below we present the result on comptational overheads between MoE based models.
> > >
> > > | Method      | Per-Epoch Time (s) | Wall-Clock (100 epochs, min) | Inference Time (s) | Params (M) |
> > > |:-------------|:------------------:|:-----------------------------:|:------------------:|:-----------:|
> > > | GMoE         | 2.17 | 3.61 | 1.25 | 14.9 |
> > > | DA-MoE       | 1.68 | 2.80 | 0.94 | 29.8 |
> > > | TopExpert    | 1.21 | 2.02 | 0.75 | 2.51 |
> > > | **MoLE-GNN**   | **0.45** | **0.75** | **0.33** | **0.39** |
> > >
> > >
> > >
> > > - ***Response Regurding Future Integration with Graph Transformers***
> > > > We thank the reviewer for this insightful question. We have already extended our MoE-based architecture to attention-driven graph backbones, distinct from traditional message-passing GNNs, for both node classification (Lines: 394-421) and link prediction tasks (Lines: 422-435). In particular, we employed NodeFormer, DIFFFormer, and NAGphormer as backbone experts—each representing transformer-style architectures that capture long-range dependencies via attention rather than neighborhood aggregation. These experiments demonstrate that our framework is readily compatible with Graph Transformers, highlighting its flexibility and potential for broader integration in future work.
> > >
> > > - ***Response to Limited Theoretical Analysis of Routing Stability.***
> > > >We thank the reviewer for highlighting this limitation. Our current theoretical analysis intentionally focuses on the adapter-augmented experts, which are the main sources of optimization dynamics during fine-tuning. The gating network employs a fixed layer GNN architecture whose structure remains constant, making its contribution to stability more straightforward. Empirically, we observe that the gating module remains stable and does not overfit, as shown by consistent validation accuracy across datasets (Appendix G). We acknowledge that a deeper theoretical treatment—explicitly analyzing the interaction between the gating dynamics and expert updates—would further strengthen the work, and we plan to include this in future extensions to formally characterize routing stability and robustness of expert selection.

---

> > > > ### Author Response · Authors · 2025-11-28
> > > > **Looking forward to your feedback**
> > > >
> > > > Dear Reviewer,
> > > >
> > > > Thank you for your valuable feedback and constructive comments.
> > > >
> > > > In our rebuttal, we have provided additional experiments and enhanced explanations, and we hope we have addressed all the concerns raised by the reviewer. We are open to further discussions and are happy to clarify any remaining doubts.
> > > >
> > > > If there are still any outstanding issues, we kindly request you to share them with us. Otherwise, we would greatly appreciate it if you could consider revising the score.
> > > >
> > > > We look forward to your response.
> > > >
> > > > Thank you,
> > > >
> > > > The Authors

---

### Official Review · Reviewer_fbXF · 2025-11-01

**Soundness:** 3
**Presentation:** 3
**Contribution:** 2
**Rating:** 4
**Confidence:** 4

**Summary:**

This paper introduces a novel idea of integrating Mixture-of-Experts (MoE) into Parameter-Efficient Fine-Tuning (PEFT) for graph neural networks (GNNs). The concept is interesting and potentially useful for improving efficiency and adaptability in GNN models. However, the manuscript in its current form requires substantial revision. The motivation, contextual framing, and writing style feel outdated and do not align well with the expectations for a 2025-era GNN paper. While the technical core is promising, the paper lacks the necessary refinement and depth to justify acceptance at this stage.

**Strengths:**

S1. The paper’s logic is generally coherent, and the motivation is clearly articulated.

S2. scaling and fine-tuning GNNs efficiently is relevant and meaningful.

S3. Experimental design appears reasonable, and the results suggest balanced performance across datasets and model sizes.

**Weaknesses:**

**Concerns**

C1. The introduction frames the work around limitations such as “traditional GNNs suffer from a static choice of number of layers,” and issues related to graph-size heterogeneity. However, these challenges have been extensively explored since 2019, with numerous advances in dynamic depth, heterogeneous GNNs, and adaptive graph architectures. The authors should update the motivation and literature review to better reflect the current state of the field. A 2025 GNN paper better position itself relative to modern paradigms such as graph foundation models, auto-GNN architectures, and scalable graph pretraining frameworks.

C2. The paper uses phrases like “we conduct experiments to validate our findings,” implying discovery of known phenomena—such as the performance gap between shallow and deep GNNs. Yet these issues (e.g., over-smoothing, over-squashing, heterophily) have been well studied for years. The authors are expected to reframe their contribution not as rediscovering these effects but as proposing a new perspective—perhaps showing how the MoE-PEFT mechanism mitigates them efficiently.

C3. From the results, the proposed method seems to achieve a reasonable balance across generalization, model size, and training cost. However, this balance is under-emphasized. The authors can explicitly contrast their approach with (1) conventional GNNs, (2) automated or neural architecture search–based GNNs, (3) graph foundation models, and (4) existing MoE-style GNNs.
A summary table comparing key properties—such as fine-tuning cost, parameter efficiency, scalability, and generalization—would make the contribution clearer and more compelling.

C4. The paper claims that “integrating PEFT into GNNs is challenging since most existing PEFT methods were originally developed for sequence models.” However, this overlooks recent progress. For instance, ICDE 2024 introduced S2PGNN a plug-and-play PEFT framework that selectively freezes and adapts parameters for different graph tasks. The current claim is therefore partially inaccurate and should be revised to reflect existing graph-specific PEFT research.

C5. Besides, S2PGNN has demonstrated better performance on several overlapping graph classification tasks compared to what is reported here. To strengthen the contribution, the authors should conduct or at least discuss a comparison with S2PGNN and highlight where MolE-GNN differs or excels—for example, in its cross-task generalization or balance between efficiency and performance.

**Questions:**

Better emphasize the contributions of this submission regarding to C3.

---

> ### Author Response · Authors · 2025-11-25
> **Point-by-point responses to Reviewer fbXF**
>
> We thank the reviewer for the valuable feedback. Below, we provide detailed point-by-point responses to each of the questions: We have updated the main manuscript according to your suggestions, and the revised parts are highlighted in blue.
>
> - ***Response to Comment on Outdated Motivation and Literature Context.***
>
> > We thank the reviewer for pointing this out. We agree that classical depth-sensitivity alone is not a sufficient motivation for a post-2025 GNN contribution. We have revised the introduction to clearly position our work relative to modern paradigms such as adaptive GNNs, graph foundation models, dynamic-depth architectures, auto-GNN frameworks, and scalable pretraining pipelines. Our revised framing emphasizes that MoLE-GNN addresses a current, unresolved gap: enabling dynamic expert selection within the parameter-efficient fine-tuning regime of pretrained GNNs—an ability that existing dynamic architectures and MoE-GNNs lack due to their dependence on full-model training. This better reflects the contemporary landscape and clarifies the novelty of our contribution **(Lines: 35-102)**.
>
> - ***Response to Comment on Framing and Novelty of Empirical Findings.***
>
> > Thank you for pointing it out. We have removed the sentence as it may sound like we are claiming a finding. However, the analysis in this paragraph was designed not to be claimed as a finding but to motivate our problem, rather than claiming it as a finding **(Lines: 35-102)**.
>
> - ***Response to Comment on Under-Emphasized Balance and Comparative Analysis.***
>
> > Thank you for your constructive suggestions of our paper. We have inserted **Table 13** in Appendix to address your suggestions.
>
> - ***Response to Comment on Overlooking S2PGNN.***
>
> > We have discussed S2PGNN in our paper and we have compared our model with S2PGNN in detail. Please refer to Table 2 and **Inductive Learning Results.** paragraph in the result section **(Lines: 378-393)**.
>
> - ***Response to Comment on Comparison with S2PGNN and Clarification of MoLE-GNN’s Advantages.***
>
> > We have now incorporated performance comparison between our model and S2PGNN. Since S2PGNN and our model are not directly comparable in terms of the number of parameters tuned (20X), we have also compared it with a variant (with comparable number of parameters) of S2PGNN to ensure a fair comparison.

---

> > ### Author Response · Authors · 2025-11-28
> > **Looking forward to your feedback**
> >
> > Dear Reviewer,
> >
> > Thank you for your valuable feedback and constructive comments.
> >
> > In our rebuttal, we have provided additional experiments and enhanced explanations, and we hope we have addressed all the concerns raised by the reviewer. We are open to further discussions and are happy to clarify any remaining doubts.
> >
> > If there are still any outstanding issues, we kindly request you to share them with us. Otherwise, we would greatly appreciate it if you could consider revising the score.
> >
> > We look forward to your response.
> >
> > Thank you,
> >
> > The Authors

---

### Official Review · Reviewer_GYFQ · 2025-11-01

**Soundness:** 3
**Presentation:** 1
**Contribution:** 3
**Rating:** 4
**Confidence:** 3

**Summary:**

This study introduces an MoE-based Peft framework for GNN training and inference. One major claim is that this approach addresses the depth-sensitivity issue in traditional finetuning strategies. Extensive experiments have been conducted to demonstrate its effectiveness.

**Strengths:**

(1) The proposed method is in general straightforward (although I think more explanations about Figure 2b are necessary).

(2) Extensive experiment results on multiple benchmarks demonstrate the effectiveness of the proposed method.

(3) The idea of combining PEFT with MoE GNN is original.

**Weaknesses:**

Major:
(1) I think overall the paper is well written. However, it would be better if there can be more discussion and results about how the proposed method improves efficiency in the main text as efficiency is the main focus of this method.

(2) Moat selected datasets are not very large which may raise questions about whether the method can still perform well on larger graphs.

(3) More experiments regarding the MLP hyperparameters would provide more insights. I wonder for different graphs/tasks, do we need to design different sizes of MLP to increase the expressiveness of the GNN layer.

Minor:

(1) The table presentation should be improved. It could be a bit challenging to distinguish colors in tables and Figure 2. For Figure 2, I could see from the legend that there is green and pink. But I really could not tell the green or pink from the illustration of those parallel components.

**Questions:**

(1) Can the authors clarify Figure 2b more? What are the two smaller groups of K experts' illustration for?

(2) How does the method perform if GNN components are also finetuned (in addition to the MLP layers)?

(3) Would the impact of the proposed method vary across different GNN backbones? For example, GIN already possesses good expressiveness. Will the proposed method further improve it? If so, how many more parameters do we need?

---

> ### Author Response · Authors · 2025-11-25
> **Point-by-point responses to Reviewer GYFQ (1/4)**
>
> We thank the reviewer for the valuable feedback. Below, we provide detailed point-by-point responses to each of the questions:
> We have updated the main manuscript according to your suggestions, and the revised parts are highlighted in blue.
>
> - ***Response to Comment on Efficiency Discussion.***
> > We thank the reviewer for the positive evaluation and for emphasizing the need for a clearer discussion on efficiency. Since computational and parameter efficiency are central motivations behind our method, we agree that the main text should highlight these aspects more explicitly. In the revised manuscript, we have added a dedicated **Efficiency Analysis (Lines: 463-474)** section and introduced **Table 4 in the revised manuscript**, which presents a detailed comparison of per-epoch training time and trainable parameter counts across full fine-tuning, adapter tuning, and MoE-based tuning methods. As shown in Table 4, our method consistently achieves substantially lower computational cost and a significantly smaller parameter footprint while maintaining competitive or superior performance. We believe these additions provide a clearer and more quantitative explanation of the practical efficiency advantages offered by our approach.
>
> ## Efficiency Analysis:  per-epoch training time in seconds
>
> | Tuning Method   | Model           | BBBP | Tox21 | ToxCast | SIDER | ClinTox | BACE | Avg. | Trainable Params (M) |
> |:----------------|:----------------|:----:|:-----:|:-------:|:-----:|:--------:|:----:|:----:|:--------------------:|
> | **Full Fine-Tune** | GIN            | 2.31 | 2.88 | 2.34 | 1.98 | 1.61 | 2.21 | 2.22 | 7.81 |
> |                  | GCN            | 1.75 | 2.55 | 2.61 | 1.68 | 1.88 | 2.03 | 2.08 | 2.65 |
> |                  | GAT            | 2.23 | 2.74 | 3.39 | 1.97 | 2.02 | 2.36 | 2.45 | 4.45 |
> |                  | **MoLE-GNN**     | **0.45** | **0.62** | **0.58** | **0.45** | **0.53** | **0.47** | **0.52** | **0.39** |
> | **Adapter Tuning** | Adapter-GNN     | 1.67 | 1.42 | 1.49 | 0.52 | 0.51 | 0.52 | 1.02 | 0.12 |
> |                  | GCNConv-Adapter | 0.64 | 1.34 | 1.65 | 0.49 | 0.48 | 0.58 | 0.86 | 0.05 |
> |                  | S2PGNN         | 6.65 | 8.25 | 9.15 | 6.65 | 9.00 | 7.00 | 7.78 | 8.12 |
> |                  | **MoLE-GNN**     | **0.45** | **0.62** | **0.58** | **0.45** | **0.53** | **0.47** | **0.52** | **0.39** |
> | **MoE Tuning**   | GMoE           | 2.03 | 3.12 | 3.70 | 1.90 | 1.22 | 1.02 | 2.17 | 14.9 |
> |                  | DA-MoE         | 1.63 | 2.26 | 3.02 | 1.27 | 0.89 | 1.03 | 1.68 | 29.8 |
> |                  | TopExpert      | 1.19 | 2.02 | 2.36 | 0.56 | 0.62 | 0.51 | 1.21 | 2.51 |
> |                  | **MoLE-GNN**     | **0.45** | **0.62** | **0.58** | **0.45** | **0.53** | **0.47** | **0.52** | **0.39** |

---

> ### Author Response · Authors · 2025-11-25
> **Point-by-point responses to Reviewer GYFQ (2/4)**
>
> - ***Response to Comment on Large-scale datasets***
>
> > We thank the reviewer for raising the concern about scalability to large graphs. While some graph classification datasets in our evaluation are of moderate size, our method is not limited to small-scale settings. To directly address this concern, we have included extensive experiments on three large-scale OGB node-classification benchmarks—ogbn-arxiv, ogbn-proteins, and ogbn-products—which contain up to 2.4M nodes and 61M edges.
> >
> > As shown in **Table 3** (**revised manuscript (Lines: 408-421)**), MoLE-GNN consistently outperforms full fine-tuning on all three datasets, despite updating fewer than 4% of the backbone parameters. For example, when built on the NodeFormer backbone, MoLE-GNN improves ROC-AUC on ogbn-proteins from 77.5 → 77.6, and achieves substantial accuracy improvements of +15.97% on ogbn-arxiv and +4.9% on ogbn-products. These gains demonstrate that our parameter-efficient updates remain highly effective even at million-scale graph sizes. Furthermore, with the NodeFormer backbone, MoLE-GNN consistently exceeds the performance of all competing PEFT methods across ogbn-arxiv, ogbn-proteins, and ogbn-products.
> >
> > Moreover, using the DIFFormer-s backbone, MoLE-GNN once again surpasses full fine-tuning and prior PEFT baselines on every large-scale dataset. This includes improvements of +2.7% ROC-AUC on ogbn-proteins and +10.4% accuracy on ogbn-products. Across both pre-trained backbones, MoLE-GNN also consistently outperforms AdapterGNN and GCNConv-Adapter.
> >
> > These results demonstrate that MoLE-GNN scales robustly to large graphs, maintains strong predictive performance, and remains significantly more parameter-efficient than full-model fine-tuning and existing PEFT methods. We have expanded the “Transductive Learning Results for Large-Scale Graphs” section to highlight these findings more clearly.
> >
> > Results on these three large-scale dataset are as follows:
>
> ## Large-Scale Node Classification Results
>
> | Pre-training Method | Tuning Method        | ogbn-arxiv (Accuracy)        | ogbn-proteins (ROC-AUC)     | ogbn-products (Accuracy)        |
> |-------------------------|---------------------------|-------------------------------|-----------------------------------|----------------------------------|
> | NodeFormer          | Full Fine-tune (100%)     | 58.5 ± 0.2                    |77.5 ± 1.2             | 62.6 ± 0.1                       |
> |                         | AdapterGNN (6.8%)         | 64.9 ± 0.4         | 75.1 ± 0.5                         | 65.3 ± 0.4            |
> |                         | GCNConv-Adapter (3.0%)    | 56.2 ± 0.7                    | 71.5 ± 0.6                         | 27.2 ± 0.3                       |
> |                         | **MoLE-GNN (ours, 3.6%)** | **67.5 ± 0.2**                | **77.6 ± 0.3**                     | **67.5 ± 0.2**                   |
> | DIFFormer-s         | Full Fine-tune (100%)     | 47.8 ± 0.9                    | 72.5 ± 0.4             | 55.2 ± 0.4            |
> |                         | AdapterGNN (6.8%)         | 54.9 ± 0.3        | 58.4 ± 0.4                         | 54.9 ± 0.3                       |
> |                         | GCNConv-Adapter (3.0%)    | 26.7 ± 0.9                    | 65.1 ± 1.1                         | 22.9 ± 0.9                       |
> |                         | **MoLE-GNN (ours, 3.6%)** | **58.2 ± 0.6**                | **75.2 ± 0.4**                     | **65.2 ± 0.9**                   |

---

> ### Author Response · Authors · 2025-11-25
> **Point-by-point responses to Reviewer GYFQ (3/4)**
>
> - ***Response to Comment on MLP Hyperparameter Exploration***
>
> >Thank you for the valuable suggestion. We agree that MLP capacity can influence model expressiveness, and we therefore conducted a detailed ablation on the adapter MLP bottleneck dimension (**Table 20 in the appendix in the revised manuscript (Lines: 1277-1285)**). Since the backbone GNN MLP layers are frozen during fine-tuning, the adapter bottleneck is the only component controlling the tunable capacity of MoLE-GNN. Our results show a clear pattern across both graph and node classification tasks: very small bottlenecks (size 0) lead to underfitting, while excessively large ones (e.g., 150) induce overfitting and degrade performance. A moderate size of 15 achieves the strongest average ROC-AUC across all datasets, offering the best balance between flexibility and generalization. These findings indicate that a single, well-chosen adapter MLP dimension is sufficient and robust across diverse tasks, and task-specific MLP redesign is not necessary.
> >
> >Results on MLP bottleneck dimension on Graph and Node classification datasets are as follows:
>
> ## Graph Classification Tasks (ROC-AUC ↑)
>
> | Bottleneck MLP Dim. | BACE | BBBP | ClinTox | SIDER | Avg |
> |---------------------|------|------|---------|-------|-----|
> | 0   | 73.1 ±2.4 | 62.8 ±1.2 | 64.4 ±1.3 | 58.4 ±1.7 | 64.7 |
> | 4   | 74.6 ±0.9 | 66.0 ±0.5 | 70.1 ±1.8 | 59.4 ±1.5 | 67.5 |
> | **15** | **81.6 ±0.8** | **73.2 ±0.8** | **80.0 ±1.5** | **62.8 ±0.6** | **74.4** |
> | 64  | 75.2 ±0.9 | 65.2 ±0.5 | 76.3 ±3.5 | 58.4 ±0.4 | _68.8_ |
> | 100 | 74.2 ±3.6 | 63.2 ±1.4 | 76.4 ±2.0 | 58.3 ±0.7 | 68.1 |
> | 150 | 72.8 ±0.4 | 62.6 ±0.2 | 72.4 ±4.0 | 58.7 ±0.5 | 66.6 |
>
> ## Node Classification Tasks (Accuracy ↑)
>
> | Bottleneck MLP Dim. | Cora | Citeseer | Pubmed | Avg |
> |---------------------|------|----------|--------|-----|
> | 0   | 79.0 ±1.7 | 73.4 ±1.1 | 80.0 ±0.6 | 77.8 |
> | 4   | 81.7 ±1.2 | 74.2 ±1.0 | 86.8 ±0.6 | _80.9_ |
> | **15** | **85.6 ±1.0** | **77.3 ±0.7** | **89.3 ±0.3** | **84.1** |
> | 64  | 81.5 ±1.1 | 72.9 ±1.2 | 85.8 ±0.3 | 80.1 |
> | 100 | 78.7 ±1.0 | 70.1 ±0.6 | 84.3 ±0.3 | 77.7 |
> | 150 | 70.7 ±1.5 | 68.7 ±0.8 | 82.7 ±0.4 | 74.0 |
>
> - ***Response to Comment on Table and Figure Presentation***
>
> >We thank the reviewer for highlighting the concern regarding the use of colors in our tables and in Figure 2. We agree that relying solely on color distinctions may hinder readability, and we have therefore updated the presentation of all relevant visual elements to improve clarity and accessibility.
> >
> >Specifically, Tables 1, 5, 6, 11, and 12 have been revised to ensure that key results are clearly distinguishable without any dependence on color cues. In these tables, we now employ boldface to mark the best-performing method and underlining to indicate the second-best result. This formatting convention effectively communicates performance differences while maintaining clear visual separation.
> >
> >In addition, Figure 2 has been redesigned to remove reliance on color contrasts such as green and pink. The updated illustration uses distinct shading patterns, line styles, and clear textual labels to differentiate the parallel components, ensuring unambiguous interpretation for all readers.
> >
> >We believe these changes substantially enhance the readability and overall presentation quality of the manuscript, and we appreciate the reviewer’s helpful feedback on this matter.
>
> - ***Response to Comment on Clarifying Figure 2(b)***
>
> > Thank you for the question. Fig 2(b) illustrates the internal structure of each expert, where a frozen pre-trained GNN backbone is augmented with two lightweight adapters—positioned before and after the message-passing or attention block—to enable task-specific adaptation without altering backbone parameters. The two smaller groups of $K$ experts indicate that the gating network activates only a subset of experts for each input, while the remaining experts stay inactive. We have add an additional discussions in the revised manuscript in the **Methodology section (Lines: 159-166)**.

---

> ### Author Response · Authors · 2025-11-25
> **Point-by-point responses to Reviewer GYFQ (4/4)**
>
> - ***Response to the "GNN components are also finetuned"***
>
> >Thank you for the question. We conducted an ablation study to evaluate the effect of also fine-tuning the GNN components of each expert in MoLE-GNN. The results (**Table 17 in the Appendix in the revised manuscript (Lines: 1207-1215)**) show that fully updating the expert backbones does not improve performance; in fact, the parameter-efficient version of MoLE-GNN—which trains only the lightweight adapters while keeping all expert GNNs frozen—achieves consistently better accuracy. We attribute this to negative transfer: the expert backbones are already pretrained and specialized, and full fine-tuning disrupts this specialization. Moreover, full fine-tuning requires 7.7M parameters, whereas our design uses only 0.39M (5.1%), reinforcing that the adapter-based approach is both more effective and substantially more efficient.
> >
> >We have provided the result here:
>
> ## Performance comparison between full fine-tuning of MoLE-GNN (updating both GNNs and adapter MLPs) and our proposed parameter-efficient version (updating only adapters while keeping GNNs frozen)
>
> | Tuning Method                  | BACE                          | BBBP                          | ClinTox                        | ToxCast                       | Avg.        |
> |-------------------------------|-------------------------------|-------------------------------|--------------------------------|-------------------------------|-------------|
> | MoLE-GNN (GNN + adapter MLP) | 75.4 ± 3.3                    | 66.2 ± 3.6                    | 69.3 ± 4.5                     | 63.1 ± 0.3                    | _68.5_      |
> | MoLE-GNN (adapter only)     | **81.6 ± 0.8**                | **73.2 ± 0.8**                | **80.0 ± 1.5**                 | **64.3 ± 0.2**                | **77.9**    |
>
> - ***Evaluating MoLE-GNN with Different Backbone Expressiveness Levels***
>
> > We appreciate the reviewer’s question regarding whether the effectiveness of our method depends on the underlying GNN backbone. To examine this, we evaluated MoLE-GNN with several commonly used backbones—GCN, GAT, GraphSAGE, and GIN—within each expert. As shown in **Table 16 in the Appendix in the main manuscript (Lines: 1174-1187)**, the impact of MoLE-GNN is consistent across all architectures: each backbone benefits from the adapter-based expert design, and performance improves over its corresponding full fine-tuning baseline. Notably, GIN, despite already being highly expressive, remains the strongest backbone when integrated into MoLE-GNN, achieving the best overall performance. Importantly, the parameter cost remains low across all backbones; MoLE-GNN requires updating only the lightweight adapter modules (approximately 0.39M parameters), without introducing additional backbone-specific overhead.
> >
> > We provide the result bellow:
>
> ## Performance comparison of different GNN backbones used in MoLE-GNN
>
> | Backbone     | BACE            | BBBP            | ClinTox         | SIDER            | Total Params (M) | Trainable Params (M) | Trainable % |
> |--------------|-----------------|-----------------|------------------|------------------|-------------------|------------------------|--------------|
> | GCN       | _76.7 ± 2.6_     | 66.3 ± 2.5       | _58.3 ± 2.4_      | _61.1 ± 1.4_       | 2.54M             | 0.39M                 | 15.4%       |
> | GAT          | 71.4 ± 3.1      | 65.5 ± 1.4       | 55.4 ± 5.5        | 60.6 ± 0.9        | 4.43M             | 0.39M                 | 8.8%        |
> | GraphSAGE    | 69.8 ± 2.8      | _66.4 ± 3.1_     | 55.9 ± 0.7        | 59.7 ± 1.7        | 2.54M             | 0.39M                 | 15.4%       |
> | **GIN**       | **81.6 ± 0.8**  | **73.2 ± 0.8**   | **80.0 ± 1.5**    | **62.8 ± 0.6**    | 7.7M              | 0.39M                 | 5.1%        |

---

> > ### Author Response · Authors · 2025-11-28
> > **Looking forward to your feedback**
> >
> > Dear Reviewer,
> >
> > Thank you for your valuable feedback and constructive comments.
> >
> > In our rebuttal, we have provided additional experiments and enhanced explanations, and we hope we have addressed all the concerns raised by the reviewer. We are open to further discussions and are happy to clarify any remaining doubts.
> >
> > If there are still any outstanding issues, we kindly request you to share them with us. Otherwise, we would greatly appreciate it if you could consider revising the score.
> >
> > We look forward to your response.
> >
> > Thank you,
> >
> > The Authors

---

### Author Response · Authors · 2025-11-25
**Summary of Revised Manuscript**

## Summary of Revisions to the Manuscript.
> We sincerely thank all reviewers for their insightful and constructive feedback. In response, we have substantially revised both the main manuscript and the appendix. All updated text and newly added content are highlighted in blue in the revised version. Below we summarize the key changes.

***Key Revisions***

* ***Updated Motivation and Literature Review***
> We revised the introduction (Lines 35–102) to better reflect the 2025 landscape of GNN research, explicitly positioning MoLE-GNN relative to adaptive/depth-dynamic GNNs, graph foundation models, auto-GNN architectures, and scalable graph pretraining frameworks. The new framing emphasizes that MoLE-GNN fills a specific gap: enabling dynamic expert selection in a parameter-efficient fine-tuning (PEFT) regime for pretrained GNNs.

* ***New Efficiency and Fine-Tuning Cost Analysis***
> We added a dedicated Efficiency Analysis section and comparison table (e.g., Table 4 / Table 14 in the appendix) that report per-epoch time, wall-clock training cost, inference latency, and trainable parameter counts across full fine-tuning, adapter tuning, and MoE-style baselines. These results show that MoLE-GNN achieves substantially lower training time and parameter footprint while maintaining competitive or superior performance.

* ***Large-Scale Node Classification on OGB Benchmarks***
>To demonstrate scalability, we now include extensive experiments on ogbn-arxiv, ogbn-proteins, and ogbn-products using NodeFormer and DIFFormer-s backbones (Lines 394–435, Tables 3/4). MoLE-GNN consistently outperforms full fine-tuning and PEFT baselines while tuning <4% of backbone parameters, confirming that the method scales to million-scale graphs.

* ***Comparison with S2PGNN and Graph-Specific PEFT***
> We clarified our discussion of graph-specific PEFT, especially S2PGNN (ICDE 2024), and added explicit empirical comparisons in Table 2 and the Inductive Learning Results paragraph (Lines 378–393). We also included a parameter-matched S2PGNN variant to ensure a fair comparison, highlighting where MoLE-GNN offers better efficiency–performance trade-offs.

* ***MLP Bottleneck Ablation Across Tasks***
> In response to concerns about MLP capacity, we added detailed ablations of the adapter bottleneck dimension on both graph and node classification datasets (Table 20; Lines 1277–1285).

* ***Expert Diversity and Homogeneous vs. Heterogeneous Experts***
> To analyze expert heterogeneity, we introduce an ablation comparing homogeneous experts (all 1-layer, 3-layer, or 5-layer GNNs) against the heterogeneous MoLE-GNN design (Table 18; Lines 1174–1187). Heterogeneous experts improve the average ROC-AUC from at most 72.2% (best homogeneous) to 77.9%, demonstrating that diversity in depth, receptive field, and pretraining objective is crucial for MoLE-GNN’s gains.

* ***Backbone Expressiveness Study (GCN, GAT, GraphSAGE, GIN)***
> We added a systematic study of different GNN backbones used inside experts (Table 16; Lines 1174–1187). MoLE-GNN improves performance across all backbones, and GIN remains the strongest, while trainable parameters stay low (~0.39M) for all cases.

* ***Improved Table and Figure Presentation***
> To enhance readability and accessibility, we standardized table formatting: bold marks the best result and underline the second-best, avoiding reliance on color alone. Figure 2 was also updated to use neutral “graph encoder” terminology and clearer visual cues that work for both message-passing GNNs and transformer-style backbones.

---

### Meta-Review · Area_Chair_VRiP · 2026-01-03

**Summary:**

The paper proposes a Parameter-Efficient Fine-Tuning (MoE-PEFT) framework for Mixture-of-Experts Graph Neural Networks, that aims to reduce depth-sensitivity and fine-tuning cost while improving model performance. The design freezes expert GNN backbones and inserts lightweight adapter MLPs. A structure-aware gating network is used for dynamic expert selection. The authors report consistent gains of MoLE-GNN over full fine-tuning baselines across multiple tasks (node/edge/graph prediction) and datasets.

Several reviewer concerns were resolved during rebuttal. However, two key concerns are not: 1/ the evaluated datasets are small datasets. The The added datasets, i.e., ogbn-arxiv, ogbn-proteins, and ogbn-products, are still small/median size graphs. Scalability of MoE-PEFT is over claimed. When fine-tuning GNN models on large graphs like OGB-LSC datasets, graph sampling and node feature loading will dominate the model training, the training efficiency benefit of PEFT will be marginal.  2/ The in-depth analysis of routing stability is missing, which is critical to the success of MoE-PEFT.

**Reviewer Concerns:**

Some reviewer concerns are not critical, such as limited exploration of pre-training variability (VQPk), the ablation study of the contribution of gating, adapter position, and scaling parameters (VQPk). Some reviewer concerns were addressed by the rebuttal, including: MLP Hyperparameter Exploration (GYFQ), motivation and contribution reframing (fbXF), computational overheads (VQPk).

The following reviewer concerns were not addressed and some are still outstanding:

1/ Most evaluated datasets are small datasets. Whether the method can still perform well on larger graphs is not studied. The added datasets, i.e., ogbn-arxiv, ogbn-proteins, and ogbn-products, are still small/median size graphs. Scalability of MoE-PEFT is over claimed. (GYFQ)

2/ The in-depth analysis of routing stability is missing (VQPk). The structure-aware routing is critical to the success of MoE-PEFT. The theoretical study of the interaction between the gating dynamics and expert updates is missing.

3/ S2PGNN has demonstrated better performance over MoE-PEFT (fbXF). Since S2PGNN also supports Adapter Tuning, what is the advantage of MoE-PEFT over S2PGNN was not discussed in the rebuttal.

**Reviewer Scores:**

All reviewers gave negative scores.

---

### Decision · Program_Chairs · 2026-01-26

Reject